# DropEdge not Foolproof: Effective Augmentation Method for Signed Graph Neural Networks

**Zeyu Zhang**[1,2], **Lu Li**[2], **Shuyan Wan**[4], **Sijie Wang**[3],
**Zhiyi Wang**[2], **Zhiyuan Lu**[2], **Dong Hao**[4], **Wanli Li**[1,2*]

[1]Engineering Research Center of Intelligent Technology for Agriculture, Ministry of Education
[2]College of Informatics, Huazhong Agricultural University, Wuhan, China
[3]The University of Auckland, [4]University of Electronic Science and Technology of China
zhangzeyu@mail.hzau.edu.cn, lu123@webmail.hzau.edu.cn,
wanshuyanzzz@163.com, swan387@aucklanduni.ac.nz,1492487431@qq.com,
1953764760@qq.com,haodong@uestc.edu.cn,liwanli@mail.hzau.edu.cn

## Abstract

Signed graphs can model friendly or antagonistic relations where edges are annotated with a positive or negative sign. Signed Graph Neural Networks (SGNNs) have been widely used for signed graph representation learning. While significant progress has been made in SGNNs research, two issues (i.e., graph sparsity and unbalanced triangles) persist in the current SGNN models. We aim to alleviate these issues through data augmentation (*DA*) techniques which have demonstrated effectiveness in improving the performance of graph neural networks. However, most graph augmentation methods are primarily aimed at graph-level and node-level tasks (e.g., graph classification and node classification) and cannot be directly applied to signed graphs due to the lack of side information (e.g., node features and label information) in available real-world signed graph datasets. Random *DropEdge* is one of the few *DA* methods that can be directly used for signed graph data augmentation, but its effectiveness is still unknown. In this paper, we are the first to provide the generalization error bound for the SGNN model and demonstrate from both experimental and theoretical perspectives that the random *DropEdge* cannot improve the performance of link sign prediction. Therefore, we propose a novel Signed Graph Augmentation framework (**SGA**) tailored for SGNNs. Specifically, SGA first integrates a structure augmentation module to detect candidate edges solely based on network information. Furthermore, SGA incorporates a novel strategy to select beneficial candidates. Finally, SGA introduces a novel data augmentation perspective to enhance the training process of SGNNs. Experiment results on six real-world datasets demonstrate that SGA effectively boosts the performance of diverse SGNN models, achieving improvements of up to 26.2% in F1-micro for SGCN on the Slashdot dataset in the link sign prediction task. Code and data are available at https://github.com/Alex-Zeyu/SGA.

## 1 Introduction

As social networks continue to gain widespread popularity, they foster a multitude of interactions among individuals, which are later documented within social graphs [1, 2]. While many of these social interactions denote positive connections, such as liking, trust, and friendship, there are also

---

[*]Corresponding Author

38th Conference on Neural Information Processing Systems (NeurIPS 2024).

instances of negative interactions, encompassing feelings of hatred, distrust, and more. For example, Slashdot [3], a tech-related news website, allows users to tag other users as either 'friends' or 'foes'. Graphs that incorporate positive and negative interactions or links are commonly termed *signed graphs* [4, 5]. In recent years, there has been a growing interest among researchers in exploring network representation within the context of signed graphs [6–8]. Most of these methods are combined with Graph Neural Networks (GNNs) [9, 10], and are therefore collectively referred to as *Signed Graph Neural Networks* (SGNNs) [11, 7, 12, 13]. This endeavor focuses on acquiring low-dimensional representations of nodes, with the ultimate goal of facilitating subsequent network analysis tasks, especially *link sign prediction*.

Despite increasing interest in SGNNs in recent years, two issues remain unresolved. First, real-world signed graph datasets are exceptionally sparse (see Table 5 from Appendix H). The sparsity of signed graphs makes downstream tasks challenging. As shown in **Issue 1** (from Figure 1), without additional structure information or side information, predicting the edge sign between nodes $v_j$ and $v_k$ in the test set is difficult. However, this changes with the introduction of extra edges ($e_{il}$) through data augmentation. Second, according to the analysis in [14], SGNNs cannot learn proper representations for nodes from unbalanced triangles. The intuitive understanding is as follows: as shown in **Issue 2**(c) (from Figure 1), there is a negative relationship between node $v_i$ and node $v_j$ in a one-hop path, while through a two-hop path via node $v_k$, it becomes a positive relationship. In other words, the relationship between node $v_i$ and node $v_j$ is uncertain, which complicates the task of SGNNs in learning representations for these three nodes. Furthermore, we notice that the proportion of unbalanced triangles is considerable (see Table 3).

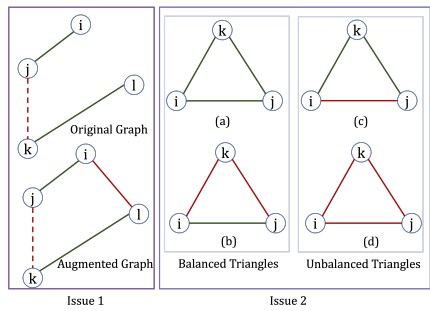

Figure 1: Green and red lines represent positive and negative edges, resp. Solid lines represent edges in the training set, while dashed lines represent edges in the test set.

One promising approach to alleviate the aforementioned issues in SGNNs is data augmentation (*DA*) which has been well studied in computer vision [15–18] and natural language processing [19–21]. Recently, significant advancements have been made in graph augmentation [22–24], including node perturbation [25, 26], edge perturbation [27], and subgraph sampling [28]. However, current graph augmentation methods cannot be directly applied to signed graphs for the following reasons: 1) Some methods [22, 23] require side information (e.g., node features and labels), which are often absent in real-world signed graph datasets that contain only structural information. 2) Random structural perturbation-based enhancement methods [27, 8, 29] cannot improve SGNN performance. As shown in Figure 2, random *EdgeDrop* cannot stably improve SGCN performance. For more experimental results on data augmentation methods based on random structural perturbations, please refer to the Appendix A. In addition, we take a first step in developing a deeper theoretical understanding of SGNN

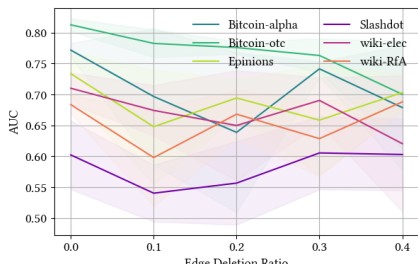

Figure 2: Effectiveness of random EdgeDrop (SGCN [11] as backbone model) on link sign prediction performance with six real-world signed graph datasets.

models and deriving the generalization error bound of the SGCN. Based on this analysis, we further demonstrate that randomly deleting edges increases the generalization error bound of SGNN, and therefore it is not an effective enhancement method for SGNN (see Section 4). In summary, it is necessary to design a new *DA* method specifically for SGNNs.

Overall, the designed signed graph augmentation method should address the two common obstacles encountered by popular SGNN models:

1. Exploring new structural information using only network information.

2. Alleviate the negative impact of unbalanced triangles on SGNNs.

To address the aforementioned obstacles, we propose a novel Signed Graph Augmentation framework, **SGA**. For the **first** obstacle, we use a classic SGNN model, such as SGCN [11] to discover candidate samples. The nodes of a signed graph are first projected into the embedding space. In the embedding space, following the principles of the extended structural balance theory [30], we interpret the relationships between closely proximate nodes as potential positive edges, whereas the relationships between more distant nodes are considered as potential negative edges. The newly added edges are treated as candidate samples (i.e., edges). To overcome the **second** obstacle, we approach it from two perspectives. *Foremost*, we attempt to reduce the number of existing unbalanced triangles. The candidate samples yield two outcomes: creating new edges or modifying the sign of existing edges. These operations do not reduce the number of training samples, which is consistent with the results of the theoretical analysis (see Sect. 4). Instead of directly incorporating the candidate samples into the training set, we analyze them beforehand. Only candidate samples that do not introduce new unbalanced triangles are retained. *Next,*, we aim to reduce the training weight of edges that belong to unbalanced triangles. To achieve this, we introduce a new perspective on data augmentation, namely edge difficulty scores (see Def. 3.3). Based on this, we design a curriculum learning training strategy tailored for SGNNs, aiming to decrease the training weight of edges with high difficulty scores and increase the training weight of edges with low difficulty scores.

To evaluate the effectiveness of SGA, we perform extensive experiments on six real-world datasets, i.e. Bitcoin-alpha, Bitcoin-otc, Epinions, Slashdot, Wiki-elec, and Wiki-RfA. We verify that our proposed SGA framework can improve the performance of the baseline models. The results of the experiment show that SGA improves the accuracy of the prediction of link signs of five base models, including two unsigned GNN models (GCN [9] and GAT [31]) and three signed GNN models (SGCN [11], SiGAT [32] and GS-GNN [13]) (see Table 1 and Table 6). SGA boosts up to 14.8% in terms of AUC for SGCN on Wiki-RfA, 26.2% in terms of F1-binary, 32.3% in terms of F1-micro, and 24.7% in terms of F1-macro for SGCN on Slashdot in link sign prediction, at best. These experiment results demonstrate the effectiveness of SGA.

**Limitations**. Our data augmentation method utilizes the conclusions from SGNN representation limitation based on balance theory [14]. However, it is well known that balance theory cannot model all signed graph formation patterns, as discussed in the paper [13]. Therefore, for real-world datasets that do not strongly conform to balance theory, our data augmentation may be less effective. Additionally, we have only validated our method on the primary downstream task of link sign prediction in signed graphs. Some works [33, 34] consider the clustering task for signed graphs, but these primarily use synthetic datasets and there are no real-world datasets available yet, which is why we have not conducted tests on them.

Overall, our contributions are summarized as follows:

- We are the first to provide the generalization error bound for the SGNN model. Based on this, we theoretically demonstrate the random *DropEdge* method, which is suitable for node classification and graph classification, is not applicable to edge-level task (i.e., link sign prediction).
- We propose a novel signed graph augmentation framework that alleviates the two issues (i.e., sparsity and unbalanced triangles) widely existing in SGNNs.
- Extensive experiments on six real-world datasets with five backbone models demonstrate the effectiveness of our framework.

## 2  Problem Statement

A *signed graph* is defined as $\mathcal{G} = (\mathcal{V}, \mathcal{E}^+, \mathcal{E}^-)$, where $\mathcal{V} = \{v_1, \ldots, v_{|\mathcal{V}|}\}$ represents the set of nodes, and $\mathcal{E}^+$ and $\mathcal{E}^-$ denote the positive and negative edges, respectively. Each edge $e_{ij} \in \mathcal{E}^+ \cup \mathcal{E}^-$ connecting two nodes $v_i$ and $v_j$ can be either positive or negative, but not both, meaning that $\mathcal{E}^+ \cap \mathcal{E}^- = \varnothing$. We use $\sigma(e_{ij}) \in \{+, -\}$ to denote the *sign* of $e_{ij}$. The structure of $\mathcal{G}$ is represented by the adjacency matrix $A \in \mathbb{R}^{|\mathcal{V}| \times |\mathcal{V}|}$, where each entry $A_{ij} \in \{1, -1, 0\}$ signifies the sign of the edge $e_{ij}$. It's important to note that, unlike unsigned graph datasets, signed graphs typically do not provide node features, meaning there is no feature vector $x_i$ associated with each node $v_i$.

*Positive* and *negative neighbors* of $v_i$ are denoted as $\mathcal{N}_i^+ = \{v_j \mid A_{ij} > 0\}$ and $\mathcal{N}_i^- = \{v_j \mid A_{ij} < 0\}$, respectively. Let $\mathcal{N}_i = \mathcal{N}_i^+ \cup \mathcal{N}_i^-$ be the set of neighbors of node $v_i$. $\mathcal{O}_3$ represents the set of

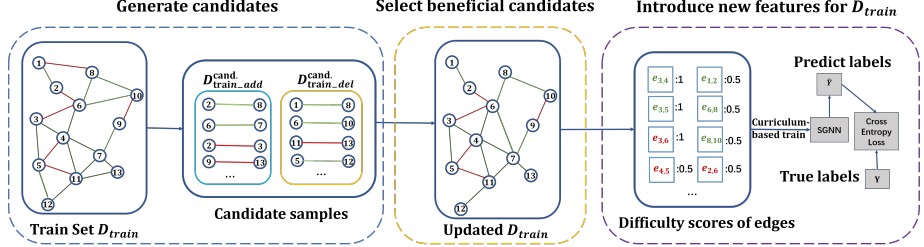

Figure 3: The overall process of SGA. Green lines represent positive edges and red lines represent negative edges.

*triangles* in the signed graph, i.e., $\mathcal{O}_3 = \{\{v_i, v_j, v_k\} \mid A_{ij}A_{jk}A_{ik} \neq 0\}$. A triangle $\{v_i, v_j, v_k\}$ is called *balanced* if $A_{ij}A_{jk}A_{ik} > 0$, and is called *unbalanced* otherwise.

**Problem Definition**: $D_{\text{train}}$ refers to the set of train samples (edges) and $D_{\text{test}}$ refers to the set of test samples. When only given $D_{\text{train}}$, our purpose is to design a graph augmentation strategy $\psi : (D_{\text{train}}) \to (D'_{\text{train}}, \mathcal{F})$, where $D'_{\text{train}}$ refers to augmented train edge set and $\mathcal{F}$ refers to the newly generated edge features (i.e., edge difficulty score).

## 3  Proposed Method

The overall framework of SGA is shown in Figure 3, which aims to augment training samples (i.e., edges) from a structure perspective (edge manipulation) to side information (edge feature). SGA mainly encompasses three key elements: 1) generating new training candidate samples, 2) selecting beneficial candidate samples, and 3) introducing a new feature (i.e., edge difficulty score) for training samples. For the specific procedural details of each part, please refer to Appendix C.

### 3.1  Generating Candidate Training Samples

Real-world signed graph datasets are extremely sparse (see Table 2) with many uncovered edges. In this part, we attempt to uncover the potential relationships between nodes. We first use a SGNN model, e.g., SGCN [11], as the encoder to project nodes from topological space to embedding space. Here, the node representations are updated by aggregating information from different types of neighbors as follows:

For the first aggregation layer $\ell = 1$:

$$
\begin{aligned}
H^{pos(1)} &= \sigma\left(\mathbf{W}^{pos(1)}\left[A^+ H^{(0)}, H^{(0)}\right]\right) \\
H^{neg(1)} &= \sigma\left(\mathbf{W}^{neg(1)}\left[A^- H^{(0)}, H^{(0)}\right]\right)
\end{aligned}
\tag{1}
$$

For the aggregation layer $\ell > 1$:

$$
\begin{aligned}
H^{pos(\ell)} &= \sigma\left(\mathbf{W}^{pos(\ell)}\left[A^+ H^{pos(\ell-1)}, A^- H^{neg(\ell-1)}, H^{pos(\ell-1)}\right]\right) \\
H^{neg(\ell)} &= \sigma\left(\mathbf{W}^{neg(\ell)}\left[A^+ H^{neg(\ell-1)}, A^- H^{pos(\ell-1)}, H^{neg(\ell-1)}\right]\right),
\end{aligned}
\tag{2}
$$

where $H^{pos(\ell)}(H^{neg(\ell)})$ represents the positive (negative) segment of representation matrix at the $\ell$th layer. $A^+(A^-)$ represents the row normalized matrix of the positive (negative) part of the adjacency matrix $A$. $\mathbf{W}^{pos(\ell)}(\mathbf{W}^{neg(\ell)})$ denotes learnable parameters of the positive (negative) part, and $\sigma(\cdot)$ is the activation function ReLU. $[.]$ is the concatenation operation. After conducting message-passing for $L$ layers, the final node representation matrix is $Z = H^{(L)} = \left[H^{pos(L)}, H^{neg(L)}\right]$. For node $v_i$, the node embedding is $Z_i$. As we wish to classify whether a pair of nodes has a positive, negative, or no edge between them. We train a multinomial logistic regression classifier (MLG) (as in [11]). The training loss is as follows:

$$\mathcal{L}\left(\theta^{\text{MLG}}\right) = -\frac{1}{|D_{\text{train}}|} \sum_{(v_i, v_j, \sigma(e_{ij})) \in D_{\text{train}}} \log \frac{\exp\left([Z_i, Z_j]\, \theta^{\text{MLG}}_{\sigma(e_{ij})}\right)}{\Sigma_{q \in \{+,-,?\}} \exp\left([Z_i, Z_j]\, \theta^{\text{MLG}}_q\right)} \tag{3}$$

$\theta^{\text{MLG}}$ refers to the parameter of the MLG classifier. Using this classifier, for any two nodes $v_i, v_j$, we can calculate the probability of forming a positive or negative edge between any two nodes, denoted as $Pr^{pos}_{e_{ij}}$ and $Pr^{neg}_{e_{ij}}$. We set up four probability threshold hyper-parameters, namely the probability threshold for adding positive edges ($\epsilon^+_{add}$), the probability threshold for adding negative edges ($\epsilon^-_{add}$), the probability threshold for deleting positive edges ($\epsilon^+_{del}$) and the probability threshold for deleting negative edges ($\epsilon^-_{del}$). Subsequently, we employ the following strategy to generate candidate training samples:

- $\forall v_i, v_j \in \mathcal{V}$, if $Pr^{pos}_{e_{ij}} > \epsilon^+_{add} \lor Pr^{neg}_{e_{ij}} > \epsilon^-_{add}$, then $D^{\text{cand.}}_{\text{train\_add}} \cup \{(v_i, v_j, \sigma(e_{ij}))\}$;
- $\forall v_i, v_j \in \mathcal{V}$, if $(v_i, v_j, \sigma(e_{ij})) \in D_{\text{train}}$, $A_{ij} > 0$, $Pr^{pos}_{e_{ij}} < \epsilon^+_{del}$, then $D^{\text{cand.}}_{\text{train\_del}} \cup \{(v_i, v_j, \sigma(e_{ij}))\}$;
- $\forall v_i, v_j \in \mathcal{V}$, if $(v_i, v_j, \sigma(e_{ij})) \in D_{\text{train}}$, $A_{ij} < 0$, $Pr^{neg}_{e_{ij}} < \epsilon^-_{del}$, then $D^{\text{cand.}}_{\text{train\_del}} \cup \{(v_i, v_j, \sigma(e_{ij}))\}$.

$D^{\text{cand.}}_{\text{train\_add}}$ and $D^{\text{cand.}}_{\text{train\_del}}$ refer to the candidate training set for adding edges and the candidate training set for deleting edges.

## 3.2 Selecting Beneficial Candidate Training Samples

After obtaining candidates, we do not merge them into the training set directly. Instead, we select the beneficial portions based on some rules. According to [14], it proves SGNNs cannot learn proper representations from unbalanced triangles (see Figure 1). The intuitive insight from this conclusion for signed graph augmentation is that beneficial candidates should not lead to new unbalanced triangles. Therefore, after generating candidate training set $D^{\text{cand.}}_{\text{train\_add}}$ and $D^{\text{cand.}}_{\text{train\_del}}$, we need to discern which operations are beneficial or not. Considering that removing edges will not introduce new unbalanced triangles, it can be directly applied to the training set. However, adding edges may potentially introduce unbalanced triangles, so it needs to be analyzed whether it should be applied to the training set. The specific criteria are as follows:

- $\forall\, (v_i, v_j, \sigma(e_{ij})) \in D^{\text{cand.}}_{\text{train\_del}}$, $D_{\text{train}} \setminus \{(v_i, v_j, \sigma(e_{ij}))\}$;
- $\forall\, (v_i, v_j, \sigma(e_{ij})) \in D^{\text{cand.}}_{\text{train\_add}}$, if $(v_i, v_j, \sigma(e_{ij})) \notin D_{\text{train}}$, and $\forall v_k \in \mathcal{N}_i \cap \mathcal{N}_j$, $(\sigma(e_{ij}) * \sigma(e_{ik}) * \sigma(e_{jk})) > 0$, then $D_{\text{train}} \cup \{(v_i, v_j, \sigma(e_{ij}))\}$.

According to the above steps, we have merged the $D^{\text{cand.}}_{\text{train\_add}}$ and $D^{\text{cand.}}_{\text{train\_del}}$ into the training set $D_{\text{train}}$.

## 3.3 Introducing New Feature for Training Samples

In this part, we attempt to alleviate the negative impact of unbalanced triangles on SGNNs from another perspective by augmenting a new feature for training samples (i.e., edges), namely edge difficulty score. Intuitively, edges belonging to unbalanced triangles have higher difficulty scores, while those belonging to balanced triangles have lower difficulty scores. Based on the difficulty scores, we design a curriculum learning-based training plan, aiming to reduce the training weights of edges with high difficulty scores, thereby mitigating the negative impact of unbalanced triangles on SGNNs.

We provide a definition of global and local balance degree:

**Definition 3.1.** The *Global Balance Degree* [35] of a signed graph is defined by:

$$D_3(\mathcal{G}) = \frac{|\mathcal{O}_3^+|}{|\mathcal{O}_3|} \tag{4}$$

where $\mathcal{O}_3$ represents the set of triangles, $\mathcal{O}_3^+$ represents the set of balanced triangles. $|\cdot|$ represents the set cardinal number.

**Definition 3.2** (Local Balance Degree). For edge $e_{ij}$, the local balance degree is defined by:

$$D_3(e_{ij}) = \frac{|\mathcal{O}_3^+(e_{ij})| - |\mathcal{O}_3^-(e_{ij})|}{|\mathcal{O}_3^+(e_{ij})| + |\mathcal{O}_3^-(e_{ij})|} \tag{5}$$

where $\mathcal{O}_3^+(e_{ij})$ ($\mathcal{O}_3^-(e_{ij})$) represents the set of balanced (unbalanced) triangles containing edge $e_{ij}$. $|\cdot|$ represents the set cardinal number.

From Def. 3.2, we can observe that the edge's local balance degree is related to the count of balanced and unbalanced triangles that include this edge. Based on this, we can define the edge difficulty score:

**Definition 3.3** (Edge Difficulty Score). For edge $e_{ij}$, the difficulty score is defined by:

$$\text{Score}(e_{ij}) = \frac{1 - D_3(e_{ij})}{2} \tag{6}$$

where $D_3(e_{ij})$ denotes the local balance degree of edge $v_{ij}$.

Upon quantifying the difficulty scores for each edge within the training set, a curriculum-based training approach is applied to enhance the performance of the SGNN model. This curriculum is fashioned following the principles outlined in [36], which enables the creation of a structured progression from easy to difficult. The process entails initially sorting the training set $\mathcal{E}$ in ascending order based on their respective difficulty scores. Subsequently, a pacing function $g(t)$ is used to allocate these edges to distinct training epochs, transitioning from easier to more challenging samples, where $t$ signifies the $t$-th epoch. we use a linear pacing function as shown below:

$$g(t) = \min\left(1, \lambda_0 + (1 - \lambda_0) * \frac{t}{T}\right) \tag{7}$$

$\lambda_0$ denotes the initial proportion of the easiest examples available, and $T$ indicates the training epoch at which $g(t)$ reaches value 1. The process of SGA is detailed in Appendix D.

# 4   Generalization Bound of SGNN

In this section, we are going to prove the generalization error bound for SGNN. Our results show that the generalization performance of the model is affected by the number of edges. A larger number of edges in the training set usually generalizes better, which means that dropout cannot always contribute to improving the model's generalization ability in many situations. For the basic setup and assumptions, please see the Appendix E.

**Main Result**. Under link prediction task, we denote $A_D$ as a learning algorithm trained on dataset $D$. According to Algorithm 2, we can set $A_D = \sigma(f(z_i, z_j, w))$, the generalization gap is defined as the difference between training error and test error:

$$\delta_{gen} = E_A[R(A_{D_{train}}) - R(A_{D_{test}})] \tag{8}$$

$$\delta_{gen} \leq \Psi\left(\beta, \theta, \frac{1}{n_t}, L\right) \tag{9}$$

**Theorem 1** (**Generalization Gap of SGNN**).

$$\delta_{gen} \leq 2\alpha_L^x + \frac{\sqrt{2}\alpha_L^y M\beta(\theta + t\eta\alpha_L^x\alpha_f\beta)}{n_t} \tag{10}$$

Here $\beta$ refers to the infinite norm of the matrix $Z$, $\theta$ refers to paradigm of initial weight matrix $\|w_{init}\|$. $R()$ is the error function and $L$ is the loss function. The generalization ability is mainly influenced by scale of the graph(the number of nodes and edges) and the norm of weights matrix. In the main result, $t$ is the number of iterations in training, $eta$ is the learning rate, $\alpha_L$, $\alpha_f$, $M$ are constants determined by the non-linear activation function, function $g$ and function $f$ respectively.

# 5   Experiments

In this section, we commence by assessing the enhancements brought about by SGA in comparison to diverse backbone models for the link sign prediction task. We will answer the following questions:

- **Q1**: Can SGA improve the performance of backbone models? Does SGA effectively alleviate issues related to graph sparsity and the presence of unbalanced triangles?

Table 1: Link sign prediction results (average $\pm$ standard deviation) with AUC (%) and F1-binary (%) on six benchmark datasets.

| Datasets | Bitcoin-alpha | | Bitcoin-otc | | Epinions | | Slashdot | | Wiki-elec | | Wiki-RfA | |
|---|---|---|---|---|---|---|---|---|---|---|---|---|
| Methods | AUC | F1-binary | AUC | F1-binary | AUC | F1-binary | AUC | F1-binary | AUC | F1-binary | AUC | F1-binary |
| GCN [9] | 60.9±0.8 | 73.6±1.6 | 69.2±0.8 | 83.0±1.5 | 68.5±0.2 | 80.5±0.1 | 51.9±0.5 | 54.7±1.0 | 64.2±0.9 | 76.6±0.7 | 60.5±0.6 | 73.3±0.4 |
| +SGA | 65.1±2.9 | 78.7±3.9 | 67.8±1.2 | 86.8±1.7 | 68.8±0.3 | 80.6±0.8 | 51.2±1.7 | 62.2±9.8 | 66.9±0.8 | 77.5±0.6 | 63.5±0.9 | 74.7±0.9 |
| (Improv.) | 6.9%↑ | 6.9%↑ | -2%↓ | 4.6%↑ | 0.4%↑ | 0.1%↑ | -1.3%↓ | 13.7%↑ | 4.2%↑ | 1.2%↑ | 5.0%↑ | 1.9%↑ |
| GAT [31] | 60.3±2.2 | 63.3±9.4 | 68.2±1.2 | 86.0±3.4 | 53.1±1.7 | 68.1±18.9 | 51.2±1.8 | 65.7±20.7 | 54.7±2.1 | 66.7±13.4 | 51.8±1.1 | 72.2±4.6 |
| +SGA | 63.0±4.5 | 86.9±2.5 | 71.7±1.4 | 90.0±3.4 | 61.4±3.7 | 80.8±6.6 | 55.2±2.1 | 68.6±11.9 | 58.5±2.0 | 72.4±4.9 | 53.4±0.6 | 69.9±3.7 |
| (Improv.) | 4.5%↑ | 37.3%↑ | 5.1%↑ | 4.7%↑ | 15.6%↑ | 18.7%↑ | 7.8%↑ | 4.4%↑ | 7.0%↑ | 8.6%↑ | 3.1%↑ | -3.2%↓ |
| SGCN [11] | 75.3±0.2 | 90.5±0.8 | 79.4±1.5 | 92.3±1.2 | 68.6±4.4 | 90.5±1.4 | 61.0±1.6 | 67.3±3.3 | 70.2±3.1 | 81.4±1.9 | 65.8±2.8 | 72.0±4.1 |
| +SGA | 80.9±2.0 | 92.8±0.7 | 82.1±0.3 | 94.6±0.3 | 77.4±0.4 | 92.2±0.9 | 68.7±1.6 | 85.0±1.0 | 77.4±0.9 | 87.0±0.7 | 75.6±0.6 | 85.5±0.8 |
| (Improv.) | 7.5%↑ | 2.5%↑ | 3.4%↑ | 2.5%↑ | 12.9%↑ | 1.9%↑ | 12.6%↑ | 26.2%↑ | 10.3%↑ | 6.9%↑ | 14.8%↑ | 18.7%↑ |
| SiGAT [32] | 85.5±0.9 | 96.8±0.1 | 88.3±1.0 | 95.5±0.2 | 89.1±0.5 | 95.2±0.1 | 84.6±0.1 | 89.2±0.1 | 88.0±0.2 | 90.9±0.1 | 87.1±0.1 | 90.3±0.0 |
| +SGA | 87.8±0.9 | 96.9±0.1 | 90.2±0.5 | 95.7±0.1 | 91.1±0.2 | 95.4±0.1 | 85.5±0.2 | 89.4±0.1 | 89.3±0.2 | 91.1±0.1 | 88.0±0.1 | 90.4±0.2 |
| (Improv.) | 2.7%↑ | 0.2%↑ | 2.1%↑ | 0.1%↑ | 2.3%↑ | 0.1%↑ | 1%↑ | 0.2%↑ | 1.5%↑ | 0.1%↑ | 1%↑ | 0.1%↑ |
| GSGNN [13] | 85.6±1.4 | 97.1±0.1 | 88.3±1.1 | 95.9±0.3 | 88.8±0.4 | 95.0±0.6 | 77.9±0.7 | 88.6±0.3 | 88.2±0.2 | 90.9±0.1 | 86.8±0.2 | 90.3±0.2 |
| +SGA | 90.0±0.1 | 97.2±0.2 | 90.7±1.2 | 96.1±0.2 | 89.6±0.5 | 95.4±0.1 | 81.2±0.2 | 88.3±0.3 | 88.8±0.2 | 91.0±0.1 | 87.5±0.1 | 90.4±0.1 |
| (Improv.) | 5.1%↑ | 0.1%↑ | 2.7%↑ | 0.2%↑ | 0.9%↑ | 0.4%↑ | 4.3%↑ | -0.3%↓ | 0.7%↑ | 0.1%↑ | 0.8%↑ | 0.2%↑ |

Table 2: Density of original graph and after augmentation.

| Dataset | Bitcoin-alpha | Bitcoin-otc | Epinions | Slashdot | Wiki-elec | Wiki-RfA |
|---|---|---|---|---|---|---|
| Original | 1.45e-3 | 8.92e-4 | 4.81e-5 | 7.55e-5 | 1.89e-3 | 1.29e-3 |
| +SGA | 1.59e-3 | 9.26e-4 | 7.29e-5 | 1.16e-4 | 3.41e-3 | 1.81e-3 |

- **Q2**: Does each part of the SGA framework play a positive role?

- **Q3**: Is the proposed method sensitive to hyper-parameters? How do key hyper-parameters impact the method performance?

For an introduction and statistical information on the datasets, please refer to the Appendix H. For details on baselines and experimental settings, please also see the Appendix I.

## 5.1 Performance Evaluation (Q1)

To comprehensively evaluate the performance of our proposed SGA, we contrast it with several baseline configurations that exclude SGA integration on *link sign prediction*. For a detailed view, AUC and F1-binary score results are presented in Table 1. Further, F1-macro and F1-micro can be referenced in Appendix J. For each model, the mean AUC and F1-binary scores, along with their respective standard deviations, are documented. These metrics are derived from five independent runs on each dataset, utilizing distinct, non-overlapping splits: 80% of the edges are used for training, while the residual 20% serve as the test set. Additionally, the table elucidates the percentage improvement in these metrics attributable to the integration of SGA, relative to the baseline models without SGA. The results provide several insights:

- Our investigations affirm that the SGA framework serves as an effective method in augmenting the performance of both signed and unsigned graph neural networks.

- For unsigned GNN models (GCN, GAT), the SGA method can effectively enhance the predictive performance (Some metrics show an improvement of over 10%.), possibly because both signed GNN and unsigned GNN models are based on a similar message-passing mechanism.

- Concerning the signed GNN models (SGCN, SiGAT, GS-GNN), we observed that SGA significantly enhances SGCN SiGAT compared to GS-GNN. The reason for this might be that SGA primarily focuses on mitigating the impact of unbalanced triangles on the model's predictive performance. Both SGCN and SiGAT are designed based on balance theory, making them more susceptible to the influence of unbalanced triangles, whereas GS-GNN is not, and therefore, it is less affected by unbalanced triangles. This indirectly reflects that SGA indeed alleviates the negative impact caused by unbalanced triangles to some extent.

Next, we will verify whether SGA effectively addresses the two issues in signed graph representation learning based on SGNN. Regarding the dataset density, we conducted a statistical analysis, as shown in Table 2. From the statistical results, it can be observed that after augmentation through the SGA method, the density of all six real-world datasets has increased, indicating a certain improvement in data sparsity issues.

Table 3: The balance degree of original datasets and after augmentation. BT refers to balanced triangle, UT refers to unbalanced triangle, BD refers to balance degree (see Def. 3.1).

| Dataset | Original | | | +SGA | | |
|---|---|---|---|---|---|---|
| | # BT | # UT | BD (%) | # BT | # UT | BD (%) |
| Bitcoin-alpha | 52,126 | 6,971 | 88.20 | 63,535 | 5,060 | 92.58 |
| Bitcoin-otc | 75,460 | 9,292 | 89.04 | 72,105 | 9,289 | 88.60 |
| Epinions | 6,286,597 | 516,723 | 92.40 | 6,162,877 | 391,707 | 94.02 |
| Slashdot | 709,417 | 64,190 | 91.70 | 676,378 | 45,268 | 93.72 |
| Wiki-elec | 311,251 | 92,934 | 77.01 | 242,691 | 29,579 | 89.13 |
| Wiki-RfA | 603,753 | 195,532 | 75.54 | 494,458 | 84,840 | 85.38 |

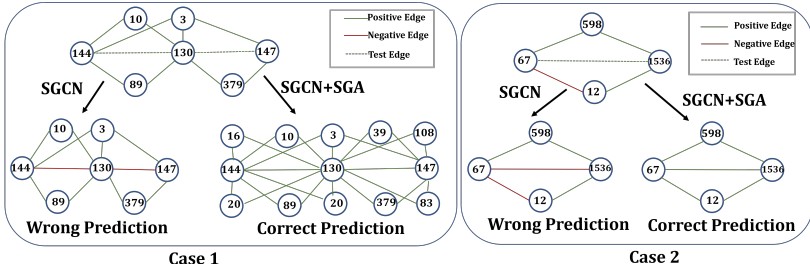

Figure 4: Case Study of SGA. Note that green lines denote positive edges and red lines denote negative edges.

Regarding unbalanced triangles, we conduct a statistical analysis, as shown in Table 3. The calculation of balance degree is based on Definition 3.1. From the statistical results, it can be seen that in most cases, SGA indeed reduces the number of unbalanced triangles, improving the balance degree of real-world datasets.

In addition to the statistical results, we also validated the effectiveness of SGA through the case study illustrated in Figure 4. Both of these two cases are from Bitcoin-alpha dataset. Case 1 verifies that SGA, by exploring latent structures, assists SGCN in correctly predicting the sign of edge $(e_{144,130}, e_{130,147})$ that are originally mispredicted. Case 2 indicates that SGA, by changing the signs of edges $(e_{67,12})$ in existing structures, reduces the impact of unbalanced triangles, allowing the model to achieve correct prediction results $(e_{67,1536})$.

## 5.2 Ablation Study (Q2)

In this section, we explore how different parts of SGA, contribute to its overall effectiveness. We do this by testing the SGCN [11] under various settings:

- **SGCN:** Here, we use the SGCN in its basic form. It works directly on the original graph data without any additional techniques or modifications.
- **+SA (Structure Augmentation, refer to Sec. 3.1 and Sec. 3.2):** SGCN operates on augmented datasets. This augmentation involves the addition or removal of edges from the initial graph.
- **+TP (Training Plan, refer to Sec. 3.3):** SGCN runs on the original graph but with a modified training paradigm. Adopting a curriculum learning approach, we rank edges by their difficulty. The model is then progressively exposed to these edges, transitioning from simpler to more challenging ones as training epochs progress.
- **+SGA(Combining both the structural augmentation and the tailored training plan)** The SGCN runs on augmented graph using a curriculum learning training plan.

Our thorough ablation study, detailed in Table 4 and conducted across six benchmark datasets, reveals several insights:

- **Importance of Structural Augmentation:** This strategy proves crucial for improving model performance. In almost all cases, using only structural augmentation leads to better results than the baseline model, which is trained on the unmodified graph without any specific training strategy.

Table 4: The ablation study results of using different components of SGA.

| Dataset | Metric | SGCN | +SA | +TP | +SGA |
|---|---|---|---|---|---|
| Bitcoin-alpha | AUC | 75.3±0.2 | 79.8±1.1 | 75.3±2.8 | **80.9±2.0** |
| | F1-binary | 90.5±0.8 | **93.8±0.4** | 91.3±1.2 | 92.8±0.7 |
| | F1-micro | 83.4±1.3 | **88.8±0.7** | 84.6±1.9 | 87.2±1.0 |
| | F1-macro | 62.1±1.1 | **69.0±1.0** | 63.1±1.7 | 67.6±0.8 |
| Bitcoin-otc | AUC | 79.4±1.5 | 80.7±2.4 | 79.7±1.0 | **82.1±0.3** |
| | F1-binary | 92.3±1.2 | 94.5±0.7 | 91.3±1.7 | **94.6±0.3** |
| | F1-micro | 86.7±1.9 | 90.2±1.2 | 86.7±2.7 | **90.5±0.4** |
| | F1-macro | 72.0±1.5 | 76.5±1.5 | 72.2±3.0 | **77.3±0.6** |
| Epinions | AUC | 68.6±4.4 | 75.9±1.0 | 75.2±1.1 | **77.4±0.4** |
| | F1-binary | 90.5±1.4 | 90.4±1.6 | 87.5±3.5 | **92.2±0.9** |
| | F1-micro | 83.9±2.0 | 84.1±2.4 | 80.1±4.7 | **86.9±1.3** |
| | F1-macro | 68.0±2.9 | 72.5±2.6 | 69.1±3.5 | **75.6±1.3** |
| Slashdot | AUC | 61.0±1.6 | 67.0±1.3 | 63.7±0.3 | **68.7±1.6** |
| | F1-binary | 67.3±3.3 | **85.3±1.5** | 67.3±1.0 | 85.0±1.0 |
| | F1-micro | 58.2±3.1 | **77.1±1.8** | 58.8±0.8 | 77.0±1.0 |
| | F1-macro | 54.6±2.4 | 67.1±1.0 | 55.8±0.6 | **68.1±0.8** |
| Wiki-elec | AUC | 70.2±3.1 | **78.0±0.5** | 71.2±2.3 | 77.4±0.9 |
| | F1-binary | 81.4±1.9 | 86.5±0.9 | 81.5±2.7 | **87.0±0.7** |
| | F1-micro | 73.1±1.7 | 80.0±0.1 | 73.4±3.0 | **80.6±0.8** |
| | F1-macro | 66.1±1.2 | 74.1±0.7 | 66.8±2.0 | **74.2±0.3** |
| Wiki-RfA | AUC | 65.8±2.7 | **75.6±0.7** | 69.8±1.2 | 75.6±0.6 |
| | F1-binary | 72.0±4.1 | 84.9±1.9 | 78.6±1.6 | **85.5±0.8** |
| | F1-micro | 63.2±3.7 | 77.8±2.2 | 70.1±1.6 | **78.6±0.9** |
| | F1-macro | 58.7±2.4 | 71.7±1.5 | 64.4±1.1 | **72.1±0.6** |

- **Effect of the Training Plan Alone:** Implementing just the training plan, without other modifications, yields a smaller performance improvement compared to using structural augmentation alone.

- **Combined Advantages of Training Plan and Data Augmentation:** Combining the training plan with structural augmentation often enhances the benefits of each approach, yielding the largest performance gain on most of the datasets.

## 5.3 Parameter Sensitivity Analysis (Q3)

In this subsection, we perform a sensitivity analysis focusing on six hyper-parameters: $\epsilon_{del}^{+}$, $\epsilon_{del}^{-}$, $\epsilon_{add}^{+}$, $\epsilon_{add}^{-}$ (these denote the probability thresholds for adding or removing positive/negative edges); $T$ represents the number of intervals during the training process where more challenging edges are incrementally added to the training set; and $\lambda_0$ designates the initial fraction of the easiest examples. Performance metrics for the SGCN model within the SGA framework, as measured by AUC across various hyper-parameter configurations, is illustrated in Figures 5. F1-binary, F1-macro and F1-micro scores are illustrated in Appendix K.

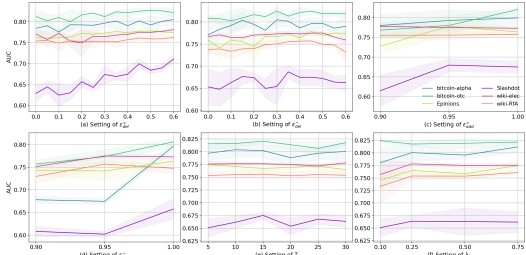

Figure 5: Performance of SGCN: AUC scores (with standard deviation) across six benchmark datasets, evaluated under variations in parameters $\epsilon_{del}^{+}$, $\epsilon_{del}^{-}$, $\epsilon_{add}^{+}$, $\epsilon_{add}^{-}$, $T$ and $\lambda_0$.

The figures reveal divergent patterns in AUC and F1 scores based on hyperparameter adjustments, with SGCN showing more significant variations on the Slashdot dataset compared to others. On a broader scale, the AUC is fairly consistent with changes to $\epsilon_{del}^{+}$ and $\epsilon_{add}^{-}$. Notably, as $\epsilon_{del}^{-}$ or $\epsilon_{add}^{+}$ rise, there's a tendency for the AUC to augment. Interestingly, AUC initially increases and then experiences a slight dip as $\lambda_0$ rises. Regarding the F1 score, it is less sensitive to changes in $\epsilon_{add}^{-}$, $T$, and $\lambda_0$, except for the case of the Slashdot dataset. In general, an increase in $\epsilon_{del}^{-}$ and $\epsilon_{add}^{+}$ boosts the F1 score. However, for $\epsilon_{del}^{+}$, the optimal value can differ across datasets, typically lying between 0.1 and 0.4.

## 6 Conclusion

In this paper, we introduce the Signed Graph Augmentation framework (SGA), a novel approach designed to address two prominent issues in existing signed graph neural networks, namely, data

sparsity and unbalanced triangles. This framework has three main components: generating candidate training samples, selecting beneficial candidate training samples, and introducing a new feature (edge difficulty score) for training samples. Based on this new feature, we have designed a curriculum learning framework tailored for SGNNs. Through extensive experiments on benchmark datasets, our SGA framework proves its effectiveness in boosting various models.

## Acknowledgments

This work was supported by the Fundamental Research Funds for the Central Universities, China [Project 2662023XXQD003, 2662023XXQD002]. We express our gratitude to Xingyu Ji and Jiale Liu, undergraduate students from the Class of 2022 at the Information College of Huazhong Agricultural University, for their assistance in conducting additional experiments during the rebuttal phase. We thank the high-performance computing platform at the National Key Laboratory of Crop Genetic Improvement in Huazhong Agricultural University.

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

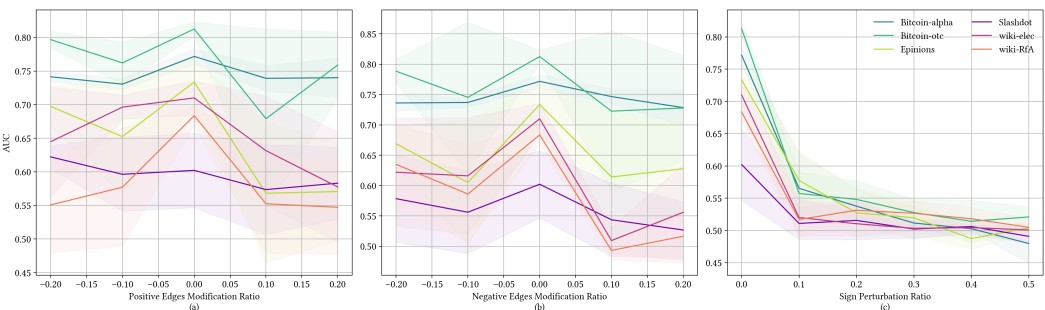

Figure 6: Effectiveness of data augmentation through random structural perturbations (SGCN [11] as backbone model) on link sign prediction performance. (a) Randomly increasing or decreasing positive edges. (b) Randomly increasing or decreasing negative edges. (c) Randomly flipping the sign of edges.

## A  Experimental results of data augmentation through random structural perturbations

We employ three different random methods (random addition/removal of positive edges, random addition/removal of negative edges, and random sign-flipping of existing edges) were tested with the classic SGNN model, SGCN [11]. As is Shown in Figure 6, The results indicate that these random methods do not enhance SGCN performance, suggesting that data augmentation methods used in signed graph contrastive learning models [8, 29] do not readily extend to general SGNNs not using a contrastive learning paradigm.

## B  Related Work

In this section, we will introduce two aspects related to this paper: signed graph neural networks and graph data augmentation.

### B.1  Signed Graph Neural Networks

Due to the widespread popularity of social media ,signed networks have become ubiquitous .Therefore, the network representation of signed graphs has gained significant attention[37, 38, 29, 14, 39, 40]. Existing research has predominantly concentrated on tasks related to *link sign prediction*, while overlooking other crucial tasks like node classification [41], node ranking [42], and community detection [43]. Some signed graph embedding techniques, such as SNE [44], SIDE [45], SGDN [46], and ROSE [47], utilize random walks and linear probability methods to capture the positive and negative relationships within graphs.These techniques consider complex interactions between nodes when processing graph data, but they may not be sufficient to capture deep-seated relationships in signed graphs. With the further exploration of signed graphs, neural networks have also been applied to signed graph representation learning in recent years. There are some SGNN models based on GCN [9].For example,the first Signed Graph Neural Network (SGNN), SGCN [11], generalizes GCN to signed graphs by utilizing balance theory to correctly aggregate and propagate the information across layers of a signed GCN model. Another noteworthy GCN-based approach is GS-GNN, which moves beyond the traditional balance theory by categorizing nodes into multiple groups. Additionally, there are SGNN models based on GAT [48] such as SiGAT [32], SNEA [7], SDGNN [12], and SGCL [8],which further enhance the ability to identify the varying levels of importance of nodes in the graph by introducing attention mechanisms. Unlike the above methods dedicated to developing more advanced SGNN models, we introduce a plugin to enhance the performance of SGNNs.

### B.2  Graph Data Augmentation

With the rapid development of graph neural networks, there has been a growing interest and demand for graph data augmentation techniques. To address issues such as data sparsity and noise

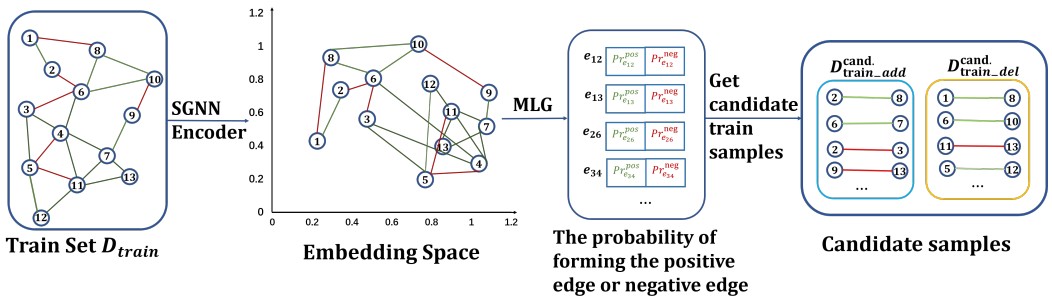

Figure 7: Generating candidate training samples

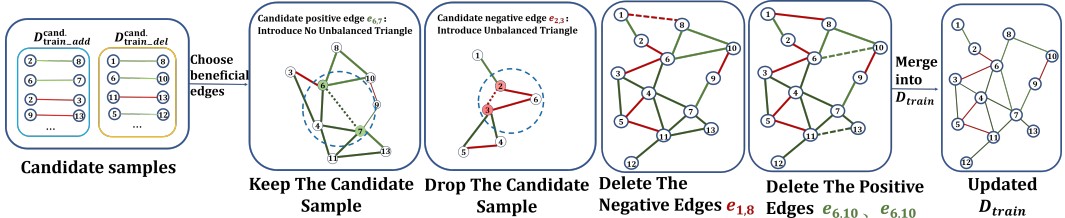

Figure 8: Selecting beneficial candidate training samples

in graphs, many research studies have focused on enhancing graph data augmentation techniques [49, 23, 24, 50–52]. According to a survey of graph data augmentation [53], graph augmentation methods can be classified into three types, i.e., feature-wise [24, 54, 55], structure-wise [56–58] and label-wise [59, 60]. Feature-wise augmentation methods primarily involve modifying, creating, or merging new features to enhance graph data. For example,LAGNN [24] operates by using a generative model that takes into account the localized neighborhood information of a target node to enrich the node's features. Other feature-wise methods [61, 25] generate augmented node features by random shuffling. Structure-wise augmentation methods target at modifying edges and nodes (e.g., randomly adding or deleting edges) to simulate different graph structures. GAUG [22] employs neural edge predictors that can effectively encode class-homophilic structure to promote intra-class edges and demote inter-class edges in a given graph structure. GraphSMOTE [62] inserts nodes to enrich the minority classes. Graph diffusion method (GDC [63]) can generate an augmented graph by providing global views of the underlying structure. Label-wise augmentation methods aim at augmenting the limited labeled training data,especially in cases where labeled data is scarce in graph data . G-Mixup [23] augment graphs for graph classification by interpolating the generator (i.e., graphon) of different classes of graphs.Note that these existing data augmentation methods rely on additional information such as node features and labels. However, the absence of key information such as node features and labels in signed graphs limits the applicability of traditional data augmentation techniques to SGNN. Therefore, it is necessary to develop data augmentation strategies tailored for signed graphs.

## C  Detailed Design of SGA

Figure 7, 8 and 9 show the details of the three steps of the SGA framework.

## D  SGA Algorithm Details

The SGA Algorithm is shown in Algorithm 1.

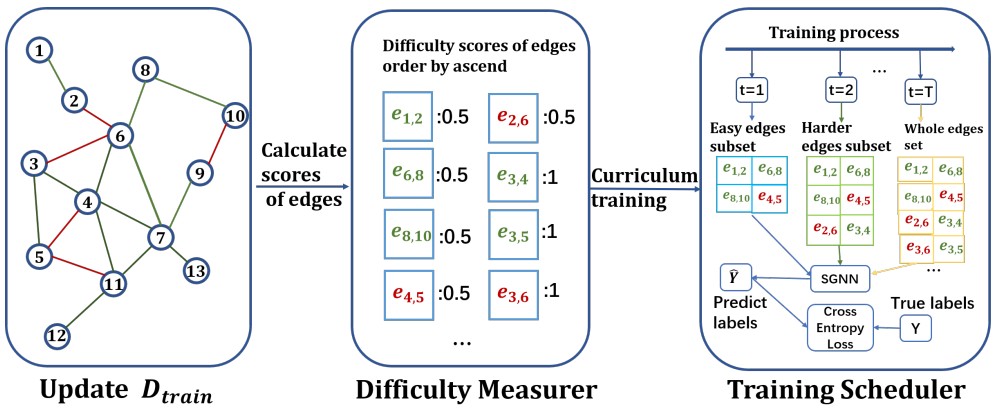

Figure 9: Introducing new feature for training samples

---

**Algorithm 1:** SGA Algorithm

---

1: **Input:** A signed graph training edge set $D_{\text{train}}$, SGCN $f$ and SGNN $f'$, pacing function $g(t)$, $\epsilon_{add}^+$, $\epsilon_{neg}^+$, $\epsilon_{add}^-$, $\epsilon_{del}^-$
2: **Output:** SGNN $f'$ parameters $\theta_{f'}$
3: Initialize SGCN parameter $\theta_f$, SGNN $\theta_{f'}$, $D_{\text{train\_add}}^{\text{cand.}} = \emptyset$, $D_{\text{train\_del}}^{\text{cand.}} = \emptyset$
4: Pre-train the $f$ on $D_{\text{train}}$
    `// Generation of Candidate Training Samples`
5: **for all** $v_i, v_j \in \mathcal{V}$ **do**
6:    Calculate $Pr_{e_{ij}}^{pos}$ and $Pr_{e_{ij}}^{neg}$ using $f$
7:    **if** $\left( Pr_{e_{ij}}^{pos} > \epsilon_{add}^+ \text{ or } Pr_{e_{ij}}^{neg} > \epsilon_{add}^- \right)$ **then**
8:        $D_{\text{train\_add}}^{\text{cand.}} \leftarrow D_{\text{train\_add}}^{\text{cand.}} \cup \{(v_i, v_j, \sigma(e_{ij}))\}$
9:    **end if**
10:   **if** $(v_i, v_j, \sigma(e_{ij})) \in D_{\text{train}}$, $A_{ij} > 0$ **and** $Pr_{e_{ij}}^{pos} < \epsilon_{del}^+$ **then**
11:       $D_{\text{train\_del}}^{\text{cand.}} \leftarrow D_{\text{train\_del}}^{\text{cand.}} \cup \{(v_i, v_j, \sigma(e_{ij}))\}$
12:   **end if**
13:   **if** $(v_i, v_j, \sigma(e_{ij})) \in D_{\text{train}}$, $A_{ij} < 0$ **and** $Pr_{e_{ij}}^{neg} < \epsilon_{del}^-$ **then**
14:       $D_{\text{train\_del}}^{\text{cand.}} \leftarrow D_{\text{train\_del}}^{\text{cand.}} \cup \{(v_i, v_j, \sigma(e_{ij}))\}$
15:   **end if**
16: **end for**
    `// Selecting Beneficial Candidate Training Samples`
17: **for all** $(v_i, v_j, \sigma(e_{ij})) \in D_{\text{train\_del}}^{\text{cand.}}$ **do**
18:   $D_{\text{train}} \leftarrow D_{\text{train}} \setminus \{(v_i, v_j, \sigma(e_{ij}))\}$
19: **end for**
20: **for all** $(v_i, v_j, \sigma(e_{ij})) \in D_{\text{train\_add}}^{\text{cand.}}$ **do**
21:   **if** $(v_i, v_j, \sigma(e_{ij})) \notin D_{\text{train}}$ **then**
22:      **for all** $v_k \in \mathcal{N}_i \cap \mathcal{N}_j$ **do**
23:         **if** $(\sigma(e_{ij})) * \sigma(e_{ik}) * \sigma(e_{jk})) > 0$ **then**
24:            $D_{\text{train}} \leftarrow D_{\text{train}} \cup \{(v_i, v_j, \sigma(e_{ij}))\}$
25:         **end if**
26:      **end for**
27:   **end if**
28: **end for**
    `// Introducing New Feature for Training Samples`
29: **for** $(v_i, v_j, \sigma(e_{ij})) \in D_{\text{train}}$ **do**
30:   Calculate $Score(e_{ij})$ using Eq. 6
31: **end for**
32: Sort $D_{train}$ according to difficulty in ascending order
33: Let $t = 1$
34: **while** Stopping condition is not met **do**
35:   $\lambda_t \leftarrow g(t)$
36:   $\mathcal{E}_t \leftarrow D_{\text{train}} [0, \ldots, \lambda_t \cdot |D_{\text{train}}|]$
37:   Use $f'$ to predict the labels $\sigma(\tilde{\mathcal{E}}_t)$
38:   Calculate cross-entropy loss $\mathcal{L}$ on $\{\sigma(\hat{\mathcal{E}}_t), \sigma(\mathcal{E}_t)\}$
39:   Back-propagation on $f$ for minimizing $\mathcal{L}$
40:   $t \leftarrow t + 1$
41: **end while**
42: **return** $\theta_{f'}$

---

# E Theory Analysis

## E.1 Basic Setup

Before the proof begins, the framework of the SGNN for edge prediction is given in Algorithm 2. A network first performs feature aggregation, where each node learns the representations of its neighboring nodes, and subsequently uses the feature representations of several pairs of nodes as inputs to a classifier for predicting the nature of the edges connecting these two nodes. Before the proof begins, we show the framework of the SGNN for edge prediction below.

---

**Algorithm 2:** Simplified SGNN Framework

---

    **Input:** the adjacency matrix $A \in \mathbb{R}^{|\mathcal{V}| \times |\mathcal{V}|}$ of graph $\mathcal{G}$, number of aggregation layers $L$, weight matrix $w$

2: **Output:** Label prediction of edge $e_{i,j}$

    $H_0 = A$

    // Feature Aggregation

4: **for** $l = 0 \rightarrow L - 1$ **do**

4:    $H_{l+1} = p(H_l, w) = \sum_{\nu_i \in \mathcal{N}^+, \nu_j \in \mathcal{N}^-} aggregate(\nu, \nu_i) + aggregate(\nu, \nu_j)$

    **end for**

6: $Z = H_L$

    // Classifier

    $\hat{\sigma}(e_{ij}) = f(z_i, z_j, w)$

8: Upgrade parameters based on loss function

    **return** $\hat{A}_{ij}$

---

A network first performs feature aggregation, where each node learns the representations of its neighboring nodes, and subsequently uses the feature representations of several pairs of nodes as inputs to a classifier for predicting the edges connecting these two nodes. Here, $z_i, z_j$ denote the i-th and j-th rows respectively, in matrix $Z$ and $\hat{A}_{ij}$ is the predicted label of edge $e_{ij}$. These vectors correspond to the node representations learned through graph filters in traditional GNNs. But actually SGNN does not have representations for each node, and this is done here to more clearly represent the variables that affect the generalization performance during the proof process next.

## E.2 Assumptions

*l*-**Lipschitz Continuous and Smooth Loss Function**: A function $f : \mathbb{X} \rightarrow \mathbb{R}$ is $\boldsymbol{\alpha}_1$-Lipschitz continuous if for all $x, y \in \mathbb{X}$, $|f(x) - f(y)| \leqslant \alpha_1 \|x - y\|$. A function $f : \mathbb{X} \rightarrow \mathbb{R}$ is $\boldsymbol{\alpha}_2$-Lipschitz smooth if for all $x, y \in \mathbb{X}$, $|f'(x) - f'(y)| \leqslant \alpha_2 \|x - y\|$, where $f'$ represents the differential function of $f$, and $\|\cdot\|$ represents the 2-norm.

Similarly, for functions of three variables, a function $f : \mathbb{X} \rightarrow \mathbb{R}$ is $\boldsymbol{\alpha}_1$-Lipschitz continuous if for all $x, y, z \in \mathbb{X}$, $|f(x_i, y_i, z) - f(x_j, y_j, z)| \leq \alpha_1 \sqrt{(x_i - x_j)^2 + (y_i - y_j)^2}$. A function $f : \mathbb{X} \rightarrow \mathbb{R}$ is $\boldsymbol{\alpha}_2$-Lipschitz smooth if for all $x, y, z \in \mathbb{X}$, $|f'(x_i, y_i, z) - f'(x_j, y_j, z)| \leq \alpha_2 \sqrt{(x_i - x_j)^2 + (y_i - y_j)^2}$. We assume that the loss function $L()$, $f()$, $g()$, satisfies the Lipschitz condition.

Actually, in simple terms, Lipschitz continuity portrays how smooth the function is, ensuring that the function does not have too steep a slope or abrupt changes: in the event of a small change in the input value, the change in the output value is also limited to a certain range.

In practice, regularization, normalization, gradient trimming, and other methods of keeping the model stable actually limit the drastic changes in the function, which means that the assumption is easily satisfied.

**Training set and Test set**: Training data is usually easier to obtain than test data. In the link prediction task, the training set contains more edges and can simulate the data distribution in real scenarios, while the test set is used to validate the performance of the model in real applications. It is also taken into account that if the number of edges in the test set is too high, it may cause the evaluation results to be skewed towards the distribution of the training set, which does not accurately

reflect the generalization ability of the model. So we assume that the training set has more edges than the test set.

# F  Proof of Theorem 1

In this section, we will present the entire proof of our main result about the generalization gap bound. Similar as proof of theorem 1[[64], Theorem 1], we have:

$$
\begin{aligned}
\delta_{gen} &= E_A\left[R\left(A_{D_{nain}}\right) - R\left(A_{D_{set}}\right)\right] \\
&= \left| \frac{1}{n_t} \sum_{e_{ij}\in\varepsilon_t^+\cup\varepsilon_t^-} L\left(\hat{A}_{ij}, A_{ij}\right) - \frac{1}{n_m} \sum_{e_{pq}\in\varepsilon_m^+\cup\varepsilon_m^-} L\left(\hat{A}_{pq}, A_{pq}\right) \right| \\
&\overset{(a)}{\leq} \frac{1}{n_t} \left| \sum_{e_{ij},e_{pq}} L\left(\hat{A}_{ij}, A_{ij}\right) - L\left(\hat{A}_{pq}, A_{pq}\right) \right|
\end{aligned}
\tag{11}
$$

The loss function we use is based on link sign prediction, where $n_t, n_m$ is the number of edges in training set and test set respectively. Step (a) is the application of our assumption. In learning process, the number of edges in test set is always lower than it in training set. Therefore, we get Equation (11). Then using absolute value inequality, we have:

$$
\begin{aligned}
&\left| \sum_{e_{ij},e_{pq}} L\left(\hat{A}_{ij}, A_{ij}\right) - L\left(\hat{A}_{pq}, A_{pq}\right) \right| \\
&\leq \sum_{e_{ij},e_{pq}} \left| L\left(\hat{A}_{ij}, A_{ij}\right) - L\left(\hat{A}_{ij}, A_{pq}\right) \right| + \left| L\left(\hat{A}_{ij}, A_{pq}\right) - L\left(\hat{A}_{pq}, A_{pq}\right) \right| \\
&\overset{(b)}{\leq} \sum_{e_{ij},e_{pq}} \alpha_L^x \left| A_{ij} - A_{pq} \right| + \alpha_L^y \left| \hat{A}_{ij} - \hat{A}_{pq} \right|
\end{aligned}
\tag{12}
$$

Step (b) uses the Lipshitz-continuity of loss function $L\left(\hat{A}, A\right)$. Next, we consider the first item. As $A_{ij}, A_{pq} \in \{1, -1, 0\}$ signifies the original sign and the predicted sign of the edge $e_{ij}$, obviously we get the upper bound: $\max|A_{ij} - A_{pq}| = 2$.

For the last item, according to the process of link prediction:

$$
\hat{A}_{ij} = \hat{\sigma}(e_{ij}) = f(z_i, z_j, w)
\tag{13}
$$

Continuing to use the Lipschitz-continuity of the function $f()$, we get:

$$
\begin{aligned}
&\sum_{e_{ij},e_{pq}} \alpha_L^y \left| \hat{A}_{ij} - \hat{A}_{pq} \right| \\
&\leq \alpha_L^y \sum_{e_{ij},e_{pq}} \left| f\left(z_i, z_j, w\right) - f\left(z_p, z_q, w\right) \right| \\
&\overset{(c)}{\leq} \alpha_L^y M \|w\| \sum_{e_{ij},e_{pq}} \sqrt{\left(z_i^2 - z_p^2\right) + \left(z_j^2 - z_q^2\right)} \\
&\leq \alpha_L^y M \|w\| \sum_{e_{ij},e_{pq}} \sqrt{z_i^2 + z_j^2} \\
&\leq \sqrt{2} \alpha_L^y M \|w\| \sum_{e_{ij},e_{pq}} \|Z\|_\infty
\end{aligned}
\tag{14}
$$

Here, Step (c) is an extension of the Lipschitz-condition for multivariate functions. Similar to $\alpha_L^y$ and $\alpha_L^x$, $M$ is constant determined only by the function $f()$ itself. We will subsequently prove the existence of the constant $M$ in the proof of Lemma 1.

In Equation (14), $\|Z\|_\infty$ refers to the infinite norm of the matrix $Z$. We set $z_i$ as the $i$-th row of matrix $Z$, which can be represented as $z_i = Z \cdot I_i$. Here $I_j = [0, \cdots, 1^j, \cdots, 0]$ is a vector with position i being 1 and the other positions being 0. The Last step followed the definition that

$$\|Z\|_\infty = \max\{\overbrace{z_1, z_2, \cdots\cdots z_i}^{|\mathcal{V}|}\}.$$

Bringing Equation (12) and (14) into Equation (11), we get the following result:

$$E_A\left[\|R(A_{D_{train}}) - R(A_{D_{test}})\|\right] \leq 2\alpha_L^x + \frac{\sqrt{2}\alpha_L^y M \|w\| \|Z\|_\infty}{n_t} \tag{15}$$

Next we will prove an upper bound for $\|w\|$ based on the SGD algorithm.

$$
\begin{aligned}
w_{t+1} &= w_t - \eta \nabla_{loss} \\
&= w_t - \eta \frac{\nabla L}{\nabla f} \cdot \frac{\nabla f}{\nabla w} \\
&\leq w_t - \eta \alpha_L^x \alpha_f |Z|_\infty
\end{aligned}
\tag{16}
$$

This gives us the iterative formula for $w$. After $t$ iterations, we can obtain upper bound of weight matrix $\|w_t\|$:

$$\|w_t\| \leq \|w_{init}\| + t\eta\alpha_L^x \alpha_f \|Z\|_\infty \tag{17}$$

Bringing Equation (17) into Equation (15), let $\beta = \|z\|_\infty$ and $\theta = \|w_{init}\|$, we get the final result:

$$\mathcal{E}_{gen} \leq 2\alpha_L^x + \frac{\sqrt{2}\alpha_L^y M\beta(\theta + t\eta\alpha_L^x\alpha_f\beta)}{n_t} \tag{18}$$

## G Proof of Lemma 1

For the ternary function $f(x, y, z)$ with a bounded $z_0$, if the function $f$ satisfies the Lipschitz-condition for $z$, it can be obtained:

$$|f(x_1, y_1, z_0) - f(x_2, y_2, z_0)| \leq M|z_0|\sqrt{(x_1 - x_2)^2 + (y_1 - y_2)^2} \tag{19}$$

For a succinct representation in the next proof, we define $D\begin{pmatrix} x_1 & x_2 \\ y_1 & y_2 \end{pmatrix} = \left(x_1 - x_2\right)^2 + \left(y_1 - y_2\right)^2$.

To prove our conclusion, we first construct the auxiliary function $G$:

$$
\begin{aligned}
&G\left(x, y, z, \Delta x, \Delta y\right) \\
&= \frac{1}{2}\left(f\left(x + \Delta x, y + \Delta y, z\right) + f\left(x, y, z\right)\right)
\end{aligned}
\tag{20}
$$

By the definition of $G(x, y, z, \Delta x, \Delta y)$, we have:

$$
\begin{aligned}
&G(x_1, y_1, z, \Delta x_1, \Delta y_1) - G(x_2, y_2, z, \Delta x_2, \Delta y_2) \\
&= \frac{1}{z}[f(x_1 + \Delta x_1, y_1 + \Delta y_1, z) - f(x_2 + \Delta x_2, y_2 + \Delta y_2, z) + f(x_1, y_1, z) - f(x_2, y_2, z)] \\
&\leq \frac{L_f}{z}\left[\sqrt{D(\begin{matrix} x_1 + \Delta x_1 & x_2 + \Delta x \\ y_1 + \Delta y_1 & y_2 + \Delta y_2 \end{matrix})} + \sqrt{D(\begin{matrix} x_1 & x_2 \\ y_1 & y_2 \end{matrix})}\right]
\end{aligned}
\tag{21}
$$

Using a special case of Cauchy Schwarz's inequality, we have:

$$D\left(\begin{array}{cc} x_1 + \Delta x_1 & x_2 + \Delta x \\ y_1 + \Delta y_1 & y_2 + \Delta y_2 \end{array}\right) \leq D\left(\begin{array}{cc} x_1 & x_2 \\ \Delta x_1 & \Delta x_2 \end{array}\right) + D\left(\begin{array}{cc} y_1 & y_2 \\ \Delta y_1 & \Delta y_2 \end{array}\right) \tag{22}$$

Given that $z$ is bounded and the obvious conclusion $\sqrt{D\left(\begin{array}{cc} x_1 & x_2 \\ y_1 & y_2 \end{array}\right)} \leq \sqrt{D\left(\begin{array}{cc} x_1 + \Delta x_1 & x_2 + \Delta x \\ y_1 + \Delta y_1 & y_2 + \Delta y_2 \end{array}\right)}$,

$$\begin{aligned} &G(x_1, y_1, z, \Delta x_1, \Delta y_1) - G(x_2, y_2, z, \Delta x_2, \Delta y_2) \\ &\leq K\sqrt{(x_1 - x_2)^2 + (y_1 - y_2)^2 + (\Delta x_1 - \Delta x_2)^2 + (\Delta y_1 - \Delta y_2)^2} \end{aligned} \tag{23}$$

Here $K$ is a constant ultimately determined by function $f$. Based on Equation (23), next we construct a differential of the following form:

$$\begin{aligned} &\frac{f(x_1, y_1, z)}{z} - \frac{f(x_2, y_2, z)}{z} \\ &= \frac{1}{|z|}\Big[f(x_1, y_1, z) + f(x_2, y_1, z) - \big(f(x_2, y_1, z) + f(x_2, y_2, z)\big)\Big] \\ &= G(x_1, y_1, z, x_2 - x_1, 0) - G(x_2, y_2, z, 0, y_2 - y_1) \end{aligned} \tag{24}$$

As what we defined in Equation (20) and (23), $\Delta x_1 = x_2 - x_1, \Delta y_2 = y_2 - y_1, \Delta x_2 = \Delta y_1 = 0$. Applying Equation (23) to Equation (24):

$$f(x_1, y_1, z) - f(x_2, y_2, z) \leq \sqrt{2}K|z|\sqrt{(x_1 - x_2)^2 + (y_1 - y_2)^2} \tag{25}$$

Let $M = \sqrt{2}K$, we managed to proof the existence of $M$ in Equation (14).

## H  Datasets

We conduct experiments on six real-world datasets, i.e., Bitcoin-OTC, Bitcoin-Alpha, Wiki-elec, Wiki-RfA, Epinions, and Slashdot. The main statistics of each dataset are summarized in Table 5. In the following, we explain the important characteristics of the datasets briefly.

**Bitcoin-OTC**[2] [65, 66] and **Bitcoin-Alpha**[3] are two datasets extracted from bitcoin trading platforms. Because Bitcoin accounts are anonymous, individuals assign trust or distrust tags to others to enhance security.

**Wiki-elec**[4] [67, 1] is a voting network in which users can choose trust or distrust to other users in administer elections. **Wiki-RfA** [68] is a more recent version of Wiki-elec.

**Epinions**[5] [67] is a consumer review site with trust and distrust relationships between users.

**Slashdot**[6] [67] is a technology-related news website in which users can tag each other as friends (trust) or enemies (distrust).

Following the experimental settings in [11], we randomly split the edges into a training set and a testing set with a ratio of 8: 2. We run with different train-test splits for 5 times to get the average scores and standard deviation.

---

[2]http://www.bitcoin-otc.com
[3]http://www.btc-alpha.com
[4]https://www.wikipedia.org
[5]http://www.epinions.com
[6]http://www.slashdot.com

Table 5: The statistics of datasets.

| Dataset | # Nodes | # Links | # Pos edges | # Neg edges | Density |
|---|---|---|---|---|---|
| Bitcoin-OTC | 5,881 | 35,592 | 32,029 | 3,563 | 1.03e-3 |
| Bitcoin-Alpha | 3,783 | 24,186 | 22,650 | 1,536 | 1.69e-3 |
| Wiki-elec | 7,115 | 103,689 | 81,345 | 22,344 | 2.05e-3 |
| Wiki-RfA | 11,017 | 170,335 | 133,330 | 37,005 | 1.40e-3 |
| Epinions | 131,580 | 840,799 | 717,129 | 123,670 | 4.86e-5 |
| Slashdot | 82,140 | 549,202 | 425,072 | 124,130 | 8.14e-5 |

## I  Baselines and Experiment Setting

We used five popular graph representation learning models as the backbones, including both unsigned GNN models and signed GNN models.

**Unsigned GNN**: We employ two classical GNN models (i.e., GCN [9] and GAT [31]). These methods are designed for unsigned graphs; thus, as mentioned before, we consider all edges as positive edges to learn node embeddings in the experiments.

**Signed Graph Neural Networks**: SGCN [11] and SiGAT [32] respectively generalize GCN [9] and GAT [31] to signed graphs based on message mechanism. In addition, they integrate the balance theory. GS-GNN [13] adopts a more general assumption (than balance theory) that nodes can be divided into multiple latent groups. We use these signed graph neural networks as baselines to explore whether **SGA** can enhance their performance.

We implement our SGA using PyTorch [69] and employ PyTorch Geometric [70] as its complementary graph library. The graph encoder, responsible for augmenting the graph, consists of a 2-layer SGCN with an embedding dimension of 64. This encoder is optimized using the Adam optimizer, set with a learning rate of 0.01 over 300 epochs. For SiGAT, we randomly standardized the embedding dimension of the node to 20 as recommended in [32]. For the remaining embedding-based methods, it was set to 64, which matches the dimensionality used in GS-GNN [13]. For the baseline methods, we adhere to the parameter configurations as recommended in their originating papers. Specifically, for unsigned baseline models like GCN and GAT, we employ the Adam optimizer, with a learning rate of 1e-2, a weight decay of 5e-4, and span the training over 500 epochs. For signed baseline models, SGCN is trained with an initial learning rate of 1e-2 and run for 300 epochs, SiGAT is trained with an initial learning rate of 5e-3 and run for 1500 epochs, and GSGNN is trained with an initial learning rate of 1e-2 and run for 3000 epochs. The experiments were performed on a Linux machine with eight 24GB NVIDIA GeForce RTX 3090 GPUs.

Our primary evaluation task is *link sign prediction*. We evaluated performance using AUC, F1-binary, F1-macro, and F1-micro metrics, consistent with established norms in related literature [71, 13]. It is imperative to note that for all these evaluation metrics, a higher score directly translates to better model performance.

## J  More Link sign prediction results

Table 6: Link sign prediction results (average $\pm$ standard deviation) with F1-micro (%) and F1-macro (%) on six benchmark datasets.

| Datasets | Bitcoin-alpha | | Bitcoin-otc | | Epinions | | Slashdot | | Wiki-elec | | Wiki-RfA | |
|---|---|---|---|---|---|---|---|---|---|---|---|---|
| Methods | F1-micro | F1-macro | F1-micro | F1-macro | F1-micro | F1-macro | F1-micro | F1-macro | F1-micro | F1-macro | F1-micro | F1-macro |
| GCN [9] | 59.9±1.9 | 45.0±0.9 | 72.9±2.0 | 57.7±1.4 | 70.6±0.1 | 60.1±0.1 | 46.3±0.6 | 44.5±0.4 | 67.0±0.7 | 60.1±0.5 | 63.0±0.3 | 56.5±0.3 |
| +SGA | 66.4±4.9 | 49.1±2.8 | 78.0±2.4 | 60.1±1.2 | 70.7±1.0 | 60.2±0.7 | 52.2±7.9 | 46.5±2.4 | 68.4±0.5 | 62.1±0.2 | 64.9±1.1 | 58.8±1.0 |
| (Improv.) | 10.9% ↑ | 9.1% ↑ | 7% ↑ | 4.2% ↑ | 0.1% ↑ | 0.2% ↑ | 12.7% ↑ | 4.5% ↑ | 2.1% ↑ | 3.3% ↑ | 3% ↑ | 4.1% ↑ |
| GAT [31] | 49.4±9.7 | 39.5±5.3 | 77.0±4.8 | 59.9±3.4 | 58.2±19.8 | 44.9±6.3 | 57.5±17.5 | 44.4±5.0 | 56.9±10.4 | 49.9±5.5 | 60.2±4.9 | 50.4±1.8 |
| +SGA | 77.5±3.7 | 53.8±2.7 | 83.0±5.0 | 65.6±4.2 | 70.7±8.4 | 55.9±1.7 | 58.7±9.2 | 51.3±4.4 | 61.8±4.9 | 54.9±3.1 | 58.3±3.4 | 50.8±1.3 |
| (Improv.) | 56.9% ↑ | 36.2% ↑ | 7.8% ↑ | 9.5% ↑ | 21.5% ↑ | 24.5% ↑ | 2.1% ↑ | 15.5% ↑ | 8.6% ↑ | 10% ↑ | -3.2% ↓ | 0.8% ↑ |
| SGCN [11] | 83.4±1.3 | 62.1±1.1 | 86.7±1.9 | 72.0±1.5 | 83.9±2.0 | 68.0±2.9 | 58.2±3.1 | 54.6±2.4 | 73.1±1.7 | 66.1±1.2 | 63.2±3.7 | 58.7±2.4 |
| +SGA | 87.2±1.0 | 67.6±0.8 | 90.5±0.4 | 77.3±0.6 | 86.9±1.3 | 75.6±1.4 | 77.0±1.0 | 68.1±0.8 | 80.6±0.8 | 74.3±0.3 | 78.6±0.9 | 72.1±0.6 |
| (Improv.) | 4.6% ↑ | 8.8% ↑ | 4.3% ↑ | 7.5% ↑ | 3.7% ↑ | 11.2% ↑ | 32.3% ↑ | 24.7% ↑ | 10.2% ↑ | 12.3% ↑ | 24.3% ↑ | 23% ↑ |
| SiGAT [32] | 94.0±0.2 | 65.2±0.7 | 91.8±0.4 | 72.4±1.3 | 91.7±0.2 | 80.7±0.5 | 82.7±0.1 | 73.0±0.1 | 85.2±0.2 | 75.8±0.5 | 84.3±0.1 | 74.4±0.2 |
| +SGA | 94.2±0.1 | 70.3±1.1 | 92.2±0.2 | 78.0±1.1 | 91.9±0.2 | 82.2±0.4 | 83.2±0.2 | 74.5±0.3 | 85.6±0.1 | 77.4±0.3 | 84.7±0.2 | 76.0±0.2 |
| (Improv.) | 0.2% ↑ | 7.8% ↑ | 0.4% ↑ | 7.7% ↑ | 0.2% ↑ | 1.9% ↑ | 0.6% ↑ | 2.1% ↑ | 0.5% ↑ | 2.1% ↑ | 0.4% ↑ | 2.1% ↑ |
| GS-GNN [13] | 94.5±0.3 | 72.1±3.7 | 93.1±1.0 | 75.4±7.5 | 91.7±1.0 | 81.9±1.7 | 81.6±0.4 | 70.3±1.0 | 85.3±0.1 | 75.8±0.6 | 84.1±0.3 | 73.0±1.1 |
| +SGA | 94.7±0.2 | 74.6±0.9 | 93.0±0.3 | 80.5±0.6 | 92.0±0.2 | 82.6±0.5 | 81.6±0.4 | 72.1±1.0 | 85.4±0.3 | 75.6±1.4 | 84.5±0.1 | 74.7±0.3 |
| (Improv.) | 0.2% ↑ | 3.4% ↑ | -0.1% ↓ | 6.7% ↑ | 0.4% ↑ | 0.8% ↑ | -0% ↓ | 2.6% ↑ | 0.1% ↑ | -0.3% ↓ | 0.5% ↑ | 2.3% ↑ |

# K  Parameter Sensitivity Analysis with F1 scores

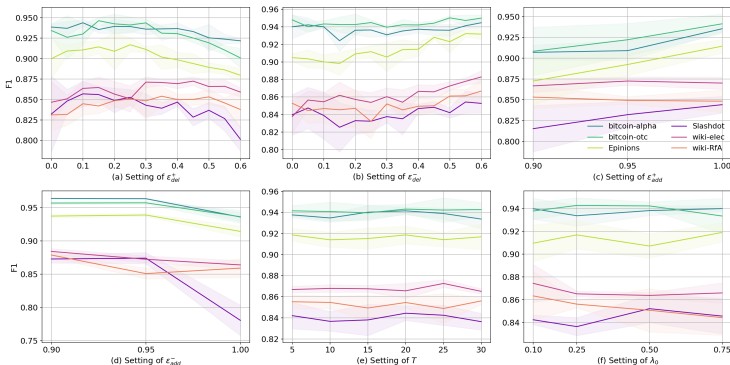

Figure 10: Performance of SGCN: F1-binary scores (with standard deviation) across six benchmark datasets, evaluated under variations in parameters $\epsilon_{del}^+$, $\epsilon_{del}^-$, $\epsilon_{add}^+$, $\epsilon_{add}^-$, $T$ and $\lambda_0$.

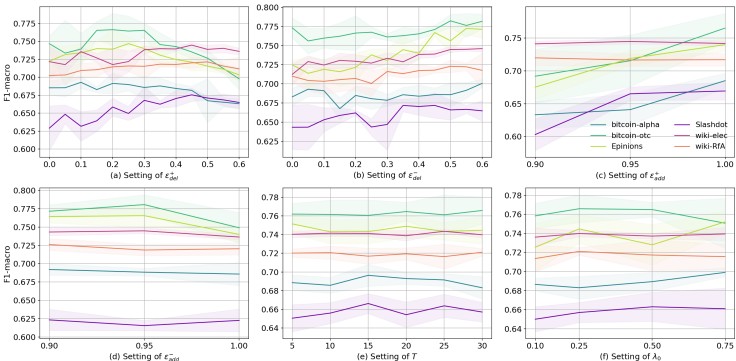

Figure 11: Performance of SGCN: F1-macro scores (with standard deviation) across six benchmark datasets, evaluated under variations in parameters $\epsilon_{del}^+$, $\epsilon_{del}^-$, $\epsilon_{add}^+$, $\epsilon_{add}^-$, $T$ and $\lambda_0$.

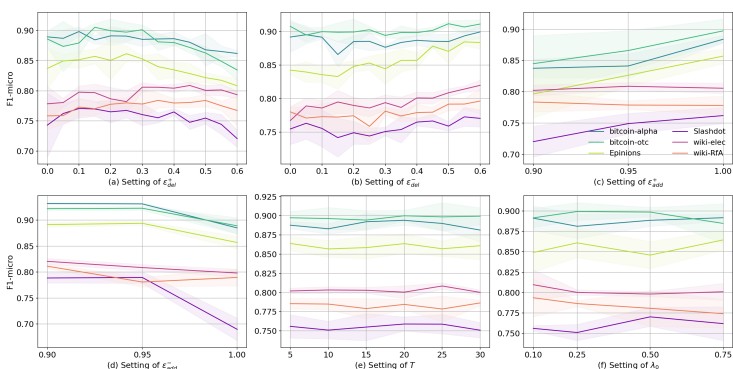

Figure 12: Performance of SGCN: F1-micro scores (with standard deviation) across six benchmark datasets, evaluated under variations in parameters $\epsilon_{del}^+$, $\epsilon_{del}^-$, $\epsilon_{add}^+$, $\epsilon_{add}^-$, $T$ and $\lambda_0$.

