# OpenReview forum: "DropEdge not Foolproof: Effective Augmentation Method for Signed Graph Neural Networks"
_NeurIPS.cc/2024/Conference — NeurIPS 2024 poster_

### Official Review · Reviewer_ShtJ · 2024-06-17

**Soundness:** 2
**Presentation:** 2
**Contribution:** 2
**Rating:** 5
**Confidence:** 4

**Summary:**

The paper presents a novel Signed Graph Augmentation (SGA) framework designed to enhance the performance of Signed Graph Neural Networks (SGNNs). The primary focus is on addressing two persistent issues in SGNNs: graph sparsity and unbalanced triangles. The authors demonstrate that the commonly used DropEdge method is ineffective for signed graph augmentation. Instead, they propose SGA, which integrates structure augmentation, candidate edge selection, and a new data augmentation perspective to improve training. Experiments on six real-world datasets show significant performance improvements in link sign prediction tasks.

**Strengths:**

Originality: The paper introduces a new augmentation framework specifically designed for SGNNs, which addresses the unique challenges of signed graphs.

Quality: The methodology is well-validated through extensive experiments on multiple real-world datasets, demonstrating substantial improvements in performance metrics.

Clarity: The paper is well-organized, with clear explanations of the proposed methods and thorough discussions of experimental results.

Significance: This work provides valuable insights and tools for improving SGNNs, which are crucial for tasks like link sign prediction in social networks.

**Weaknesses:**

Generalization: The effectiveness of SGA on other types of signed graph tasks, such as node classification or community detection, has not been explored.

Complexity: The proposed framework involves multiple steps and parameters, which may complicate its implementation and tuning.

Resource Intensive: The computational requirements for the proposed method may be high, potentially limiting its practical application in resource-constrained environments.

**Questions:**

- Can the proposed SGA framework be adapted for other signed graph tasks, such as node classification or community detection?

- How does the performance of SGA scale with larger datasets or more complex network structures?

- What are the computational overheads associated with the different components of the SGA framework, and are there ways to optimize them?

**Limitations:**

The authors acknowledge that their data augmentation method is based on balance theory, which may not be applicable to all real-world signed graph datasets. Additionally, the method has only been validated on link sign prediction tasks, and its effectiveness on other tasks remains untested. To improve, the authors could explore the generalization of SGA to other tasks and consider alternative theoretical frameworks for signed graph augmentation. Potential societal impacts, especially in contexts involving negative social interactions, should also be addressed.

---

> ### Author Rebuttal · Authors · 2024-08-07
>
> **For Weakness 1 and Q1 on more downstream tasks:**
>
> Thanks for your constructive comments. Existing SGNN methods focus primarily on the link sign prediction task, and they ignore the performance on other tasks. We experiment with the performance (metric: Average Accuracy± standard deviation) of SGA on different baselines in node classification and community detection tasks. The datasets were sourced from [16] .
>
> 1. Node Classification
>
> |model|        rainfall        |        samspon         |
> |:-:|:----------------------:|:----------------------:|
> |SGCN|    0.72 ± 0.05     |    0.76 ± 0.17     |
> |SGCN+SGA |    0.75 ± 0.03     |    0.72 ± 0.18     |
> |improv.|  **4.2% $\uparrow$**   | **-5.3% $\downarrow$** |
> |GSGNN|    0.72 ± 0.06     |    0.44 ± 0.30     |
> |GSGNN+SGA|    0.68 ± 0.02     |    0.44 ± 0.26     |
> |improv.| **-5.6% $\downarrow$** |       **0% -**   |
> |SiGAT|    0.64 ± 0.23     |    0.60 ± 0.20     |
> |SiGAT+SGA|    0.65 ± 0.13     |    0.68 ± 0.18     |
> |improv.|  **1.2% $\uparrow$**   |  **13.3% $\uparrow$**  |
>
> 2. Community Detection
>
> |model|rainfall|          ppi           |
> |:-:|:-:|:----------------------:|
> |SGCN|0.49 ± 0.15|    0.26 ± 0.09     |
> |SGCN+SGA |0.60 ± 0.04 |    0.26 ± 0.09     |
> |improv.|**22.4% $\uparrow$**|        **0% -**        |
> |GSGNN|0.39 ± 0.02|    0.36 ± 0.01     |
> |GSGNN+SGA|0.40 ± 0.01 |    0.33 ± 0.06     |
> |improv.|**2.6% $\uparrow$**| **-8.3% $\downarrow$** |
> |SiGAT|0.53 ± 0.15|    0.14 ± 0.01     |
> |SiGAT+SGA|0.60 ± 0.08 |    0.16 ± 0.02     |
> |improv.|**13.2% $\uparrow$**|  **14.3% $\uparrow$**  |
>
> Based on the experimental results, the SGA shows minimal improvement for the node classification task, but it significantly enhanced the performance for the community detection task.
>
> **For Weakness 2:**
>
> Please see Common Concern.
>
> **For Weakness 3 and Q3 on computational overheads:**
>
> 1. Model parameter statistics
> |Model|#params|
> |:-:|:-:|
> | SGCN |14851|
> |GSGNN |30580|
> |SiGAT |185600|
> |SGCN+SGA|29702|
> |GSGNN+SGA|45431|
> |SiGAT+SGA|200451|
>
> 2. model training time cost
> BitcoinOTC:
> |Model|time(s)|
> |:-:|:-:|
> |SGCN|75.4 ± 6.4|
> |GSGNN|142.8 ± 17.2|
> |SiGAT|277.0 ± 30.7|
> |SGCN+SGA|149.6 ± 42.7|
> |GSGNN+SGA|5217 ± 48.0|
> |SiGAT+SGA|351.2 ± 41.2|
>
> The method proposed maintains a relatively stable number of additional parameters and training time, even for large models, avoiding increased computational complexity. Training is efficient and can be completed quickly. The additional operations required by the method can be optimized using parallelization, divide-and-conquer strategies, and feature reduction algorithms. More detailed statistical results are available in the referenced PDF.
>
> **For Q2 on the performance on large datasets:**
>
> We tested the performance of SGA on the Amazon-CD dataset [17]. The dataset contains 895,266 edges and 97,731 nodes, making it one of the larger signed graph datasets we could find.
>
> |            | AUC              | F1-Binary  | F1-Micro   | F1-Macro   |
> |------------|------------------|------------|------------|------------|
> | SGCN       | 61.65 ± 0.14 | 69.11 ± 1.41 | 58.56 ± 1.14 | 53.04 ± 0.69 |
> | SGCN+SGA   | 58.26 ± 0.37       | 83.08 ± 1.47 | 72.47 ± 1.65 | 57.44 ± 0.39 |
> | SIGAT      | 55.08 ± 1.45       | 89.96 ± 0.29 | 81.76 ± 0.48 | 45.02 ± 0.48 |
> | SIGAT+SGA  | 64.58 ± 0.29     | 89.51 ± 0.32 |  81.83 ± 0.20  |53.28 ± 0.17   |
>
> Based on the experimental results, our method improves the performance of the baselines on the Amazon-cd dataset.
>
> **For Limitation on alternative theoretical framework:**
>
> Regarding the analysis of SGA's generalization, we have conducted the following brief discussion.
>
> SGA prevents the formation of unbalanced triangles in the input graph while selecting beneficial training samples. This stabilization of the graph structure preserves the eigenvalue distribution, thereby enhancing the quality of the graph embedding. Let $L$ be the Laplacian matrix with unbalanced triangles and $\lambda_i$ its eigenvalues. After eliminating unbalanced triangles, the Laplacian matrix and eigenvalues become $L'$ and $\lambda'_i$, with $\Delta L$ representing the matrix perturbation. The corresponding change in eigenvalues can be expressed as
>
> $$\Delta\lambda_i=\lambda_i^{'}-\lambda_i=\mathbf{v}_i^T\Delta L\mathbf{v}_i,$$
>
> Where $v_i$ the eigenvector of $L$, $Z$ the original embedding vector, and $Z'$ the embedding vector after eliminating unbalanced triangles. Removing unbalanced triangles reduces the absolute values of the Laplacian matrix's non-diagonal elements, decreasing the perturbation matrix $\Delta L$. Consequently, the change in eigenvalues $\lambda_i$ decreases, leading to $\mathrm{Var}(\lambda) > \mathrm{Var}(\lambda')$. By Spectral Graph Theory, this implies $\mathrm{Var}(Z) > \mathrm{Var}(Z')$.
>
> Let $H$ be the original embedding space and $H^{'}$ be the embedding space after variance reduction. Since the variance of the embedding vectors becomes smaller, the distribution range of the embedding vectors becomes smaller. We have:$$\|\|\mathbf{Z}_i-\bar{\mathbf{Z}}\|\|^2>\|\|\mathbf{Z}_i^{'}-\bar{\mathbf{Z}}^{'}\|\|^2.$$
>
>
> Thus, coverage number $N(\epsilon,\mathrm{H}^{\prime},\|\cdot\|)$ will be less than coverage number $N(\epsilon,\mathcal{H},\|\cdot\|)$ for the same $\epsilon$.
>
> In particular, the covering number can be used to define an upper bound on Rademacher complexity [15]. $H$ is the hypothesis space, $N(\epsilon,\mathcal{H},\|\|\cdot\|\|)$ is the number of covers, and Rademacher complexity
> $R_n(H)$ can be defined by the following inequality:
>
> $$\mathrm{R}_n(\mathrm{H})\leq2\epsilon+3\sqrt{\frac{\log N(\epsilon,\mathrm{H},\|\|\cdot\|\|)}n}.$$
>
> We've got $$\log N(\epsilon,\mathrm{H}^{'},\|\|\cdot\|\|)<\log N(\epsilon,\mathrm{H},\\||\cdot\|\|).$$
>
> Thus we finally show that SGA can be useful in improving the generalization performance by decreasing the number of coverages in the embedding space while increasing the number of samples, which ultimately shrinks the upper bound of $R_n(H)$.

---

> > ### Comment · Reviewer_ShtJ · 2024-08-13
> >
> > Thank you for your response. The additional experiments provided are comprehensive and thorough; the updated figure is clearer and more comprehensible. Based on these improvements, I am pleased to upgrade the score.

---

> > > ### Author Response · Authors · 2024-08-13
> > >
> > > Thank you for your recognition. If you have any other questions, please feel free to ask, and we will respond promptly.

---

### Official Review · Reviewer_QdQs · 2024-07-04

**Soundness:** 3
**Presentation:** 3
**Contribution:** 3
**Rating:** 6
**Confidence:** 3

**Summary:**

This work addresses the scarcity of effective data augmentation strategies tailored for signed graphs, especially considering the dearth of auxiliary information in real-world datasets. By presenting the generalization error bound for SGNNs and disproving the universal benefit of random DropEdge, this paper introduces Signed Graph Augmentation (SGA). SGA innovates with a structure augmentation module identifying potential edges from network patterns and employs a selective strategy to enrich training data. The proposed method significantly improves SGNN performance.

**Strengths:**

The paper is well structured and has clear writing. The task of signed graph augmentation is interesting and novel. The experimental results show the effectivity of the proposed method.

**Weaknesses:**

1. Regarding the experimental results shown in Table 1, the paper only shows the results of solely using the backbone model and with SGA augmentation method. The efficiency of the proposed method will be more convincing if it can be compared with another augmentation method. In addition to the DropEdge method, there are still other augmentation methods, e.g., mixup-based methods [1,2] and spectrum-based methods [3,4].
2. Though the performance when using SGA is better, compared to do not using any augmentation method, it introduces many modules to train, e.g., the encoder and the MLG classifier, thus introducing many more parameters. A comparison of a number of parameters and the training time will be better.
3. Minor comments: Figure 3 can be improved to be more clear. The Encoder, MLG classifier, and the classifier loss are not explicitly shown in the figure.

Reference:
[1] G-mixup: Graph data augmentation for graph classification. ICLR’22
[2] Graph mixup with soft alignments. ICML’23
[3] Spectral augmentation for self-supervised learning on graphs. ICLR’22
[4] Through the Dual-Prism: A Spectral Perspective on Graph Data Augmentation for Graph Classification. arXiv'24

**Questions:**

1. In Section 4, according to ‘the generalization performance of the model is affected by the number of edges’, will randomly adding edges will be more effective than dropping edges in signed graphs? Is there any empirical evidence about this?
2. Lack of complexity analysis and cost time comparison. Is SGA efficient in large-size datasets?

### After rebuttal
Most of my concerns have been addressed. I raise my score to 6.

**Limitations:**

The authors discussed the limitations of the proposed method.

---

> ### Author Rebuttal · Authors · 2024-08-07
>
> **For Weakness 1 on other augmentation methods:**
>
> Thank you for the reviewers' suggestions. We carefully reviewed the recommended papers [1-4]. Although the data augmentation methods discussed in those articles are not specifically designed for the link sign prediction task, their underlying concepts have been highly inspiring for our work. We will include a discussion of the aforementioned methods in the Related Work section of the final version. We selected one method each from the mixup-based methods and spectrum-based methods as augmentation techniques, namely S-Mixup [2] and DP-noise [4], to be applied in link sign prediction on Bitcoin-otc dataset.
>
> |Model|AUC|F1|Micro|Macro|
> |:-:|:-:|:-:|:-:|:-:|
> |SGCN|78.3 ± 1.4|90.5 ± 2.6|84.0 ± 3.9|69.2  ± 4.1|
> |SGCN+S-Mixup|76.8 ± 2.3|88.8 ± 2.9|81.5 ± 4.1|67.4 ± 3.5|
> |improv.|-1.9% $\downarrow$|-1.9% $\downarrow$|-3.0% $\downarrow$|-2.6% $\downarrow$|
> |GSGNN|89.1 ± 1.2|96.5 ± 0.7|93.6 ± 1.2|80.8 ± 2.0|
> |GSGNN+S-Mixup|87.0 ± 2.1|96.2 ± 0.2|93.1 ± 0.3|79.4 ± 1.7|
> |improv.|-2.4% $\downarrow$|-0.3% $\downarrow$|-0.5% $\downarrow$|-1.7% $\downarrow$|
> |SiGAT|76.5 ± 2.5|94.9 ± 0.2|90.4 ± 0.3|59.2 ± 1.6|
> |SiGAT+S-Mixup|57.1 ± 4.4|65.9 ± 22.2|56.2 ± 21.2|43.5 ± 12.2|
> |improv.|-25.4% $\downarrow$|-30.6% $\downarrow$|-37.8% $\downarrow$|-26.5% $\downarrow$|
>
>
> |Model|         AUC         |         F1          |        Micro        |        Macro        |
> |:-: |:-------------------:|:-------------------:|:-------------------:|:-------------------:|
> |SGCN|    84.20 ± 0.12     |    93.69 ± 0.16     |    89.07 ± 0.26     |    76.55 ± 0.35     |
> | SGCN+DP-noise |    79.26 ± 0.41     |    92.74 ± 0.48     |    87.42 ± 0.76     |    72.84 ± 0.67     |
> |improv. | -5.85% $\downarrow$ | -0.99% $\downarrow$ | -1.92% $\downarrow$ | -4.92% $\downarrow$ |
> | GSGNN |    71.19 ± 7.73     |    96.28 ± 0.60     |    93.13 ± 1.19     |    75.29 ± 8.70     |
> | GSGNN+DP-noise |    70.36 ± 6.30     |    96.03 ± 0.39     |    92.69 ± 0.81     |    74.36 ± 6.60     |
> |improv. | -1.16% $\downarrow$ | -0.26% $\downarrow$ | -0.47% $\downarrow$ | -1.23% $\downarrow$ |
> |SiGAT |    76.52 ± 1.98     |    94.85 ± 0.17     |    91.08 ± 0.31     |    59.24 ± 1.61     |
> | SiGAT+DP-noise |    85.20 ± 0.65     |    95.11 ± 0.49     |    91.10 ± 0.50     |    70.87 ± 2.80     |
> |improv.|  11.34% $\uparrow$  |  +0.00%   |  +0.00%  |  19.63% $\uparrow$  |
>
> Based on the experimental results, S-Mixup did not effectively enhance the baseline, because the fusion operation of the S-Mixup at the feature level loses the structural information of the graph. However, DP-noise demonstrated effectiveness on certain metrics, and we believe that this kind of method of preserving the graph structure is worth further exploring. We will include these experimental results in the final version.
>
> **For Weakness 2 and Q2 on parameter number and training time**:
>
> 1. Model parameter statistics
>
> |Model|#params|#change|
> |:-:|:-:|:-:|
> | SGCN |14851|-|
> |GSGNN |30580|-|
> |SiGAT |185600|-|
> |SGCN+SGA|29702|14851 $\uparrow$|
> |GSGNN+SGA|45431|14851 $\uparrow$|
> |SiGAT+SGA|200451| 14851$\uparrow$|
>
> 2. Model training cost time statistics
>
> Bitcoin-alpha:
> |Model|time(s)|change(s)|
> |:-:|:-:|:-:|
> |SGCN|51.0 ± 5.3|-|
> |GSGNN|186.8 ± 42.5|-|
> |SiGAT|257.6 ± 25.8|-|
> |SGCN+SGA|197.4 ± 74.1|146.4 $\uparrow$|
> |GSGNN+SGA|295.2 ± 65.9|108.4 $\uparrow$|
> |SiGAT+SGA|395 ± 79.6|137.4 $\uparrow$|
>
> BitcoinOTC:
> |Model|time(s)|change(s)|
> |:-:|:-:|:-:|
> |SGCN|75.4 ± 6.4|-|
> |GSGNN|142.8 ± 17.2|-|
> |SiGAT|277.0 ± 30.7|-|
> |SGCN+SGA|149.6 ± 42.7|74.2 $\uparrow$|
> |GSGNN+SGA|5217 ± 48.0|5074.2 $\uparrow$|
> |SiGAT+SGA|351.2 ± 41.2|74.2 $\uparrow$|
>
> 3. Complexity analysis please see Common Concern.
>
> Overall, the amount of additional parameters and training time for our method is relatively stable. When the model is large, our method does not bring greater computational complexity. Training can be carried out in a relatively short amount of time.
>
> **For Weakness 3 on Figure 3**:
>
> Thank you for your suggestion. We have made revisions to Figure 3 and have submitted a revised version, which is included in the uploaded PDF. Due to time constraints, we will continue making further modifications later.
>
> **For Q1 on randomly adding edges**：
>
> Due to space limitations, we only present the performance of different baselines on the bitcoin-otc dataset with a randomly increased proportion of edges. The performance on other datasets is provided in the uploaded PDF file. Please refer to the uploaded PDF for detailed statistical results across more datasets.
>
>
> SGCN:
> |add_edge|AUC|F1|Micro|Macro|
> |:-:|:-:|:-:|:-:|:-:|
> |0%|78.3 ± 1.4|90.5 ± 2.6|84.0 ± 3.9|69.2  ± 4.1|
> |5%|77.4 ± 2.5|89.1 ± 1.9|81.8 ± 2.8|66.8 ± 3.0|
> |10%|75.5 ± 1.5|85.6 ± 2.9|76.8 ± 4.0|62.5 ± 3.0|
> |15%|74.7 ± 1.6|85.4 ± 2.9|76.4 ± 4.1|61.9 ± 3.1|
> |20%|67.5 ± 9.0|64.2 ± 32.3|57.7 ± 24.4|47.3 ± 19.4|
>
> GSGNN:
> |add_edge |AUC|F1|Micro|Macro|
> |:-:|:-:|:-:|:-:|:-:|
> |0%	|89.1 ± 1.2|96.5 ± 0.7|93.6 ± 1.2|80.8 ± 2.0|
> |5%	|87.1 ± 1.6|96.1 ± 0.1|92.9 ± 0.3|78.7 ± 1.0|
> |10%|86.4 ± 1.1|96.1 ± 0.1|92.8 ± 0.2	|78.1 ± 1.1|
> |15%|84.5 ± 1.1|95.9 ± 0.2|92.5 ± 0.4|77.7 ± 0.6|
> |20%|85.2 ± 1.4|95.7 ± 0.3|92.2 ± 0.5|77.3 ± 1.4|
>
> SiGAT:
> |add_edge|AUC|F1|Micro|Macro|
> |:-:|:-:|:-:|:-:|:-:|
> | 0%| 76.5 ± 2.5| 94.9 ± 0.2|90.4 ± 0.3|59.2 ± 1.6|
> | 5% | 72.8 ± 3.7 | 94.7 ± 0.2  | 90.0 ± 0.4| 55.4 ± 4.3 |
> | 10%| 69.8 ± 5.3| 94.6 ± 0.1| 89.8 ± 0.2 | 54.7 ± 3.1  |
> | 15% | 70.1 ± 4.2 | 94.5 ± 0.1| 89.7 ± 0.2  | 54.4 ± 2.4 |
> | 20% | 70.3 ± 3.0| 94.6 ± 0.1 | 89.9 ± 0.2  | 55.2 ± 2.4 |
>
>
> Overall, randomly adding edges tends to degrade the model's performance, and the decline is more significant compared to randomly removing edges (refer to Fig.2 in the paper).

---

> > ### Comment · Reviewer_QdQs · 2024-08-12
> >
> > Thank you for your response. The additional experiments provided are comprehensive and thorough; the updated figure is clearer and more comprehensible. Based on these improvements, I am pleased to upgrade the score.

---

> > > ### Author Response · Authors · 2024-08-12
> > >
> > > Thank you for your comments and recognition of this work.
> > >
> > > If you have any other questions, please let us know and we will present timely responses.

---

### Official Review · Reviewer_B6jA · 2024-07-09

**Soundness:** 4
**Presentation:** 3
**Contribution:** 3
**Rating:** 7
**Confidence:** 5

**Summary:**

Link sign prediction is a significant downstream task in graph data analysis. Current graph data augmentation methods seldom explore this task. This paper analyzes, both theoretically and experimentally, why existing graph data augmentation methods perform poorly on this task and proposes a new data augmentation method tailored for signed graph analysis. Its main contributions include the following two aspects:

- It provides a generalization error bound for signed graph neural networks and theoretically verifies that the widely used random edgedrop method is not suitable for the link sign prediction task.
- It proposes a novel signed graph data augmentation framework to address two major issues in current signed graph neural networks (SGNNs), namely sparsity and imbalanced triads.

**Strengths:**

1. Although significant progress has been made in the field of graph augmentation, data augmentation methods specifically for signed graphs and link sign prediction are relatively new research directions. Currently, there are relatively few data augmentation methods specifically designed for edge-level tasks in graph augmentation.
2. In this data augmentation module, the authors propose an intriguing new approach, which is to treat the training difficulty of training samples as a feature for augmentation, thereby adjusting the training weights of these samples.
3. The paper is well structured, logically coherent and very easy to understand.

**Weaknesses:**

1. According to the authors' theoretical analysis, reducing the number of training samples does not contribute to improving model performance. However, in Section 3.2, we observed that some training samples were removed because they belonged to imbalanced triplets. Does this observation contradict the theoretical analysis?
2. The authors mention in the limitations section that "for real-world datasets that do not strongly conform to balance theory, our data augmentation may be less effective." What is the basis for this statement? Are there possible solutions proposed by the authors for this issue?
3. Some typographical errors: on line 164, the concatenation operation should use [,] instead of [.]. On line 225, there is a missing space between SGNN and "our results". Also, some formulas have punctuation while some do not.

**Questions:**

Please refer to Weaknesses.

**Limitations:**

The paper argues that the data augmentation method can alleviate rather than completely solve the current obstacles of SGNN methods. This claim is supported by experimental and theoretical analyses. However, a limitation of this article is that it only addresses a subset of issues within GNN models and their impact on downstream tasks, thus its influence may be somewhat limited.

---

> ### Author Rebuttal · Authors · 2024-08-07
>
> (1) **For Weakness 1 on some training samples removed :**
>
> Our theoretical analysis indeed demonstrates that reducing the number of edges during training can degrade model performance. However, the SGA method does not reduce the overall number of edges. Specifically, we take a cautious approach to edge removal by setting a high threshold (over 0.9, up to 1) before executing any edge removal operation, as detailed in our published code. As a result, the number of edges in the training set actually increases after the data augmentation process. This is corroborated by Table 2, where the graph density is shown to improve following data augmentation.
>
> (2) **For Weakness 2 on limitation part:**
>
> Regarding the issues raised in the *limitations*:
>
> Most existing SGNN methods are built on the GCN architecture, and it has been shown [6] that such models struggle to learn appropriate representations from unbalanced cycles. Our data augmentation method relies heavily on these SGNN models to identify suitable potential candidates (i.e., edges). Therefore, designing a signed graph representation framework based on alternative architectures could better address this issue. For example, leveraging large language models (LLMs) could be a promising approach. However, the integration of LLMs with graph structures is still in its early stages, and, to our knowledge, no studies have specifically targeted signed graphs. The reliability of directly applying existing frameworks to this context remains uncertain and requires further validation. Overall, this is a problem worth exploring in greater depth.
>
> (3) **For Weakness 3 on typos:**
>
> Thank you for the reviewer’s comments. We will correct the typos in the final version of the paper.
>
> (4) **For Limitations on influence:**
>
> Signed graphs, which assign positive or negative signs to edges, are powerful tools for modeling complex relationships across various fields. They offer valuable insights into social network dynamics, biological and chemical interactions, recommendation systems, and international relations. By capturing both positive and negative interactions, signed graphs facilitate a deeper understanding of intricate systems, providing sophisticated analytical methods to address challenges in multiple disciplines.

---

> > ### Comment · Reviewer_B6jA · 2024-08-13
> >
> > The author's response basically addressed my concerns. I also carefully read the author's responses to other reviewers and found them to be very comprehensive. I will stand with my rating.

---

> > > ### Author Response · Authors · 2024-08-13
> > >
> > > Thank you for your feedback and appreciation; if you have any further questions, please don't hesitate to reach out, and we'll respond promptly.

---

### Official Review · Reviewer_wfoh · 2024-07-10

**Soundness:** 3
**Presentation:** 2
**Contribution:** 3
**Rating:** 7
**Confidence:** 5

**Summary:**

This paper proposes a new research subfield focusing on data augmentation methods for signed graphs. Unlike the widely studied unsigned graph augmentation, this method targets the downstream task of link sign prediction rather than the mainstream node classification [1] or graph classification [2]. As far as I know, most current graph data augmentation methods mainly address node classification tasks, while there is relatively little enhancement work for edge tasks, or their effectiveness for link prediction is not significant [3]. Additionally, this article provides the first generalization error bound for signed graph neural networks, which is used to analyze why current commonly used data augmentation methods yield unstable results for signed graphs. In designing data augmentation methods, the article introduces a new perspective from curriculum learning.

[1] Kazi, Anees, et al. "Differentiable graph module (dgm) for graph convolutional networks." IEEE Transactions on Pattern Analysis and Machine Intelligence 45.2 (2022): 1606-1617.
[2] Chen, Yu, Lingfei Wu, and Mohammed Zaki. "Iterative deep graph learning for graph neural networks: Better and robust node embeddings." Advances in neural information processing systems 33 (2020): 19314-19326.
[3] Gasteiger, Johannes, Stefan Weißenberger, and Stephan Günnemann. "Diffusion improves graph learning." Advances in neural information processing systems 32 (2019).

**Strengths:**

- This paper proposes a new subfield of research with a wide range of applications. According to my investigation, this article is indeed the first paper on this topic.
- This paper introduces a novel approach to data augmentation, which involves using curriculum learning to adjust the training weights of challenging edges. This perspective is quite insightful.
- This paper presents the first generalization error bound for signed graph neural networks and, based on this, analyzes the reasons why current commonly used graph data augmentation methods (such as random edge deletion) yield unstable results for link sign prediction.

**Weaknesses:**

- Although the article introduces a new problem within a subfield, its application scope is relatively narrow compared to more generalized graph structures.
- Why can't current graph augmentation methods be directly applied to signed graph representation learning? In the limitations, the authors mention that for datasets with poor balance, the effectiveness of this data augmentation method decreases. Are there any good solutions to this problem?
- Can the authors provide a more detailed analysis of the factors affecting SGA's performance? Specifically, why do some unsigned and signed GNNs show significant performance enhancements on certain datasets, while others exhibit only marginal improvements or none at all?
- Some typos: In line 231, there is a missing space after "algorithm 2". In line 236, there is also a missing space after "the main result".

**Questions:**

See weaknesses.

**Limitations:**

See weaknesses.

---

> ### Author Rebuttal · Authors · 2024-08-07
>
> (1) **For weakness 1 on application scope:**
>
> Compared to more generalized graphs, signed graph analysis has its exclusive downstream task (i.e., link sign prediction) which are important and very interesting, such as product reviews [10], bill votes [11], paper reviews, polarization study [14], echo chambers [12]. In addition to social networks, the field of bioinformatics also utilizes this approach, for example, to predict upregulation and downregulation relationships between diseases and genes [13].
>
> (2) **For weakness 2 on other graph augmentation methods and limitation issues:**
>
> Regarding the current graph data augmentation methods that cannot be directly applied to signed graphs, we primarily discuss this in the 3rd paragraph of the introduction. In summary, it mainly includes two aspects:
>
> 1. Most methods are designed for node classification [5], graph classification [1-2], and link prediction tasks, with no existing approaches specifically targeting link sign prediction. Moreover, these methods rely on side information, such as node features and labels, which are often missing in most real-world signed graph datasets that contain only structural information [7-8].
>
> 2. Random structural perturbation based augmentation methods [15] cannot improve SGNN performance. We have verified this from both experimental and theoretical perspectives. (see Fig.2 and Sec.4)
>
> Regarding the issues raised in the **limitations**:
>
> Given that most current SGNN methods are designed based on the GCN architecture, and This paper [6] has demonstrated that current SGNN models based on such architectures are unable to learn suitable representations from unbalanced cycles. Our data augmentation method heavily relies on this kind of SGNN models to mine suitable potential candidates (i.e., edges), so designing a signed graph representation encoding framework based on alternative architectures could better address the current issue. For instance, using large language models to tackle this problem could be a promising approach. However, the integration of large language models with graphs is still relatively limited. According to our research, there is currently no studies specifically targeting signed graphs. Whether it is reliable to directly use some of the existing frameworks also needs further validation. Overall, this is a very worthwhile problem to explore.
>
> (3) **For weakness 3 on Experimental Results:**
>
> Here are **two observations** regarding the experimental results:
>
> 1. The performance improvement of the methods based on balance theory architectures, namely SGCN [7] and SiGAT [8], is more significant compared to GS-GNN [9]. In other words, the improvements achieved by SGCN and SiGAT are more noticeable, whereas the improvements with GS-GNN are relatively smaller.
>
> 2. The overall performance on the first four datasets (Bitcoin-alpha, Bitcoin-otc, Epinion, Slashdot) is better than on the last two datasets (Wiki-elec, Wiki-Rfa).
>
> Regarding Observation 1, from the perspective of method design, the information fusion mechanism of the model itself influences the performance of SGA. We believe the reasons might include two aspects. First, as indicated by the analysis in RSGNN [6], the current SGNN methods based on balanced theory [7] (i.e., SGCN, SiGAT) fail to learn appropriate representations from unbalanced cycles, whereas SGA effectively reduces the proportion of unbalanced cycles (see Table 3), leading to better enhancement for these two methods. Second, since GS-GNN [9] is not limited to the balanced theory assumption, it can handle unbalanced cycles well, thus the enhancement effect from SGA is less significant.
>
> Regarding Observation 2, from the perspective of dataset balance, we believe that the performance of SGA is related to the balance of the dataset. As shown in Table 3, the initial balance degree (BD %) of the first four datasets is better than that of the last two datasets. Considering that we use SGCN as the link sign prediction model to identify potential candidates, its prediction performance is poorer for datasets with low balance degree. As evidenced in Table 1, this conclusion holds true. Therefore, for datasets with low balance, it is more challenging to identify suitable potential candidates, leading to a decrease in the overall data augmentation effectiveness.
>
> (4) **For weakness 4 on typos:**
>
> Thank you for the reviewer’s comments. We will carefully review the paper and correct typos in the final version.

---

> > ### Comment · Reviewer_wfoh · 2024-08-13
> >
> > Thanks for the author's response. It has largely addressed my questions and provided a better understanding of the details of the paper. Therefore, I will keep my score unchanged.

---

### Author Rebuttal · Authors · 2024-08-07

Thank all reviewers for their valuable and constructive comments. We address the common concern here and believe the quality of the paper has been improved following the reviewers' suggestions.

**Common Concern: Time and Space Complexity of SGA**

Suppose we are given a signed graph $\mathcal{G}=(\mathcal{V},\mathcal{E}^+,\mathcal{E}^- )$, where $\mathcal{V} = ({v_1, \ldots, v_{|\mathcal{V}|}})$  represent the set of nodes, $\mathcal{E}^+$  and $\mathcal{E}^-$ respectively denote the positive and negative edges, and the structure of $\mathcal{G}$ is represented by the adjacency matrix $A∈\mathbb{R}^{|\mathcal{V}|×|\mathcal{V}|}$.

**Background**

The computation of the l-th layer of a SGNN network is:
$$
X^{L+1}=σ(AX^l W^l)
                             =σ(A^+ X^l W^l+A^- X^l W^l)
$$

Where $σ()$ is a non-linear activation function, $W^l$ is a feature transformation matrix $∈\mathbb{R}^{F_l×F_{l+1}}$, and $A^+$ and $A^-$ are the adjacency matrix of positive edges and negative edges, respectively. For simplicity, we assume the node features at every layer are size-F. As such, $W^l$ is an $F×F$ matrix.

**Time Complexity Analysis:**

1) SGNN

     We analyze the complexity of the SGNN by three high-level operations:

	i. feature transformation ($Z^L=X^l W^l$),

	ii. neighborhood aggregation ($A^+ Z^L,A^- Z^L$),

	iii. activation function ($σ()$).

Part i. is a dense matrix multiplication between matrices of size $N×F_l$ and $F_l×F_{l+1}$. As our assumption $F_l=F_{l+1}=F$, the complexity is $O(NF^2)$.

Part ii. is a multiplication between matrices of size $N×N$ and $N×F$, yielding $O(N^2 F)$. We use a sparse operator, neighborhood aggregation for each node therefore requires $O(d^+ F+d^- F)$, where $d^+$ and $d^-$ respectively represent the positive and negative neighbors of each node on average. Finally, we have a total complexity of $O(NdF)$, where $d=d^++d^-$.

Part iii. is simply an element-wise function, so its cost is $O(N)$.

Over $L$ layers, the complexity is $O(L×(NF^2+NdF+N)$ for one forward propagation.
The time complexity of backpropagation is usually the same as that of forward propagation because it requires the gradient to be computed and the parameters to be updated is the same as forward propagation.

2) SGA

	We decompose the complexity of the SGA by two operations:

	i. Updating training data by similarity calculation,

	ii. Training model by curriculum learning.

Part i. involves a dense matrix multiplication between two matrices of size $N×F$, yielding $O(N^2 F)$. Note that, this part runs only once and does not participate in forward and backward propagation.  In our experiments, this process can be optimized in parallel, and its time complexity is lower than that of SGN.

Part ii. is an upgraded version of the original SGNN. Similar to the analysis in section 1, we can have a complexity of $O(L×(NF^2+NdF+N))$ in this part.

Finally, we conclude that the complexity of SGA is the same as that of SGNN, which is $O(L×(NF^2+NdF+N))$.

**Space Complexity Analysis:**

1) SGNN

	i. The space complexity of the input feature matrix X is $O(NF)$.

	ii. The adjacency matrix A: $O(N^2)$ which can be optimized by sparse operator $O(Nd)$.

	iii. The space complexity of the output feature matrix $H^{l+1}$ is $O(NF)$, where $l$ represents the current layer and $l+1$ represents the next layer.

	iV. The space complexity of the weight matrix W is $O(F^2 )$.

The space complexity of SGNN is $O(Nd+NF+F^2 )$ for one layer network.

2) SGA


	i. The space complexity of the input feature matrix X is $O(NF)$.

	ii. The adjacency matrix A: $O(Nd+M)$ where M is the augmented edges and $M<Nd$.

	iii. The space complexity of the output feature matrix $H^{l+1}$ is $O(NF)$, where $l$ represents the current layer and $l+1$ represents the next layer.

	iV. The space complexity of the weight matrix W is $O(F^2 )$.

In the case of using the same activation function, the space complexity of SGA is also $O(Nd+NF+F^2 )$.

**Experimental Setup**:  Our experimental results are reported as the mean and standard deviation calculated over 5 runs. The uploaded PDF includes supplementary experimental results and an updated framework figure.

**References throughout the entire rebuttal:**

[1] Han, Xiaotian, et al. G-mixup: Graph data augmentation for graph classification. ICML'22.

[2] Ling, Hongyi, et al. Graph mixup with soft alignments. ICML'23.

[3] Lin, Lu, et al. Spectral augmentation for self-supervised learning on graphs. ICLR'22

[4] Xia, Yutong, et al. Through the Dual-Prism: A Spectral Perspective on Graph Data Augmentation for Graph Classification. arXiv'2024.

[5] Zhao, Tong, et al. Data augmentation for graph neural networks. AAAI'21.

[6] Zhang, Zeyu, et al. Rsgnn: A model-agnostic approach for enhancing the robustness of signed graph neural networks. WWW'23.

[7] Derr, Tyler, Signed graph convolutional networks. ICDM'18.

[8] Huang, Junjie, et al. Signed graph attention networks.ICANN'19

[9] Liu, Haoxin, et al. Signed graph neural network with latent groups. KDD'21.

[10] Seo, Changwon, et al. SiReN: Sign-aware recommendation using graph neural networks. TNNLS'23.

[11] Huang, Junjie, et al. Signed bipartite graph neural networks. CIKM'21.

[12] Tzeng, Ruo-Chun, et al. Discovering conflicting groups in signed networks. Neurips'20.

[13] Zhang, Guangzhan, et al. SGNNMD: signed graph neural network for predicting deregulation
types of miRNA-disease associations. BIBM'22.

[14] Xiao, Han, et al. Searching for polarization in signed graphs: a local spectral approach. WWW'20.

[15] Tang, Huayi, et al. Towards understanding generalization of graph neural networks. ICML'23.

[16] He, Yixuan, et al. SSSNET: semi-supervised signed network clustering. SDM'22.

[17] Chen, Sirui, et al. SIGformer: Sign-aware Graph Transformer for Recommendation. SIGIR'24.

---

### Decision · Program_Chairs · 2024-09-25

**Decision:**

Accept (poster)

**Comment:**

The paper studies the problem of data augmentation in signed GNNs (SGNNs). Specifically, it first derives generalization bounds for SGNNs, while it also introduces a new augmentation strategy to improve upon DropEdge's performance on link sign prediction. The proposed approach has merits that the reviewers have highlighted. These mostly include the novelty of the methodology as well as the theoretical and empirical support.  The authors have adequately addressed most of the reviewers' concerns. Therefore, I recommend accepting the paper.